# RANDOM-SET NEURAL NETWORKS

**Shireen Kudukkil Manchingal**[1][*]        **Muhammad Mubashar**[1]        **Kaizheng Wang**[2, 4]

**Keivan Shariatmadar**[3,4]        **Fabio Cuzzolin**[1]

[1]School of Engineering, Computing and Mathematics, Oxford Brookes University, UK
[2]M-Group and DistriNet Division, Department of Computer Science, KU Leuven, Belgium
[3]LMSD Division, Mechanical Engineering, KU Leuven   [4]Flanders Make@KU Leuven
{19185895, 19230664, fabio.cuzzolin}@brookes.ac.uk
{kaizheng.wang, keivan.shariatmadar}@kuleuven.be

## ABSTRACT

Machine learning is increasingly deployed in safety-critical domains where erroneous predictions may lead to potentially catastrophic consequences, highlighting the need for learning systems to be aware of how confident they are in their own predictions: in other words, 'to know when they do not know'. In this paper, we propose a novel Random-Set Neural Network (RS-NN) approach to classification which predicts *belief functions* (rather than classical probability vectors) over the class list using the mathematics of *random sets*, i.e., distributions over the collection of *sets* of classes. RS-NN encodes the 'epistemic' uncertainty induced by training sets that are insufficiently representative or limited in size via the size of the convex set of probability vectors associated with a predicted belief function. Our approach outperforms state-of-the-art Bayesian and Ensemble methods in terms of accuracy, uncertainty estimation and out-of-distribution (OoD) detection on multiple benchmarks (CIFAR-10 vs SVHN/Intel-Image, MNIST vs FM-NIST/KMNIST, ImageNet vs ImageNet-O). RS-NN also scales up effectively to large-scale architectures (e.g. WideResNet-28-10, VGG16, Inception V3, EfficientNetB2 and ViT-Base-16), exhibits remarkable robustness to adversarial attacks and can provide statistical guarantees in a conformal learning setting.

## 1 INTRODUCTION

Machine learning models often struggle to provide reliable predictions when confronted with unfamiliar data (Guo et al., 2017a; Ovadia et al., 2019; Minderer et al., 2021), may it be noisy samples (Papernot et al., 2016) deliberately designed to deceive models, or out-of-distribution data (OoD) beyond the model's training distribution. An ideal learning system, in opposition, should be aware of how confident it is in its own predictions, acknowledge the limits of its knowledge and gauge these limitations to make informed decisions by modelling the *epistemic uncertainty* associated with it, in essence, 'to know when it does not know'. Epistemic uncertainty pertains to uncertainty associated with our ignorance about the underlying data generation process (Kendall & Gal, 2017; Hüllermeier & Waegeman, 2021; Manchingal & Cuzzolin, 2022). Within machine learning, a major source of epistemic uncertainty arises from the limited representativeness of the available training data, constrained in both quantity and quality.

In this paper, we propose a new approach to classification which models epistemic uncertainty using a *random set* approach (Molchanov, 2005; 2017). As they assign probability values to sets of outcomes directly, random sets can naturally model the fact that observations often come in the form of sets (in particular, when missing data occurs), and accommodate ambiguity, incomplete data, and non-probabilistic uncertainties. As classification involves only a finite list of classes, we model uncertainty using *belief functions* (Shafer, 1976), the finite incarnation of random sets (Cuzzolin, 2018), whose theory (Shafer, 1976) is, in fact, a generalisation of Bayesian inference (Smets, 1986). Classical (discrete) probabilities can be seen as special belief functions, and Bayes' rule as a special case of the Dempster's rule of combination (Dempster, 2008) originally proposed for aggregating belief functions. For readers less familiar with the topic, in §A.1 we recall the distinction between classical probabilities and belief functions and the way they handle uncertainty in more detail. Fig.

---

[*]Corresponding author.

1 contrasts the inference processes of our *Random-Set Neural Network* (RS-NN) and of a Bayesian Neural Network (Jospin et al., 2022). In hierarchical Bayesian inference (top), a posterior distribution over the network's weights is learned from a training set. At prediction time, a predictive distribution is generated in the target space (left) by sampling from this weight posterior (middle), which amounts to a second-order probability distribution there (Hüllermeier & Waegeman, 2021). A single (mean) prediction is then typically derived by Bayesian Model Averaging (BMA) (Hoeting et al., 1999) (right), while uncertainty is measured by the entropy of the mean prediction and the variance of the predictive distribution.

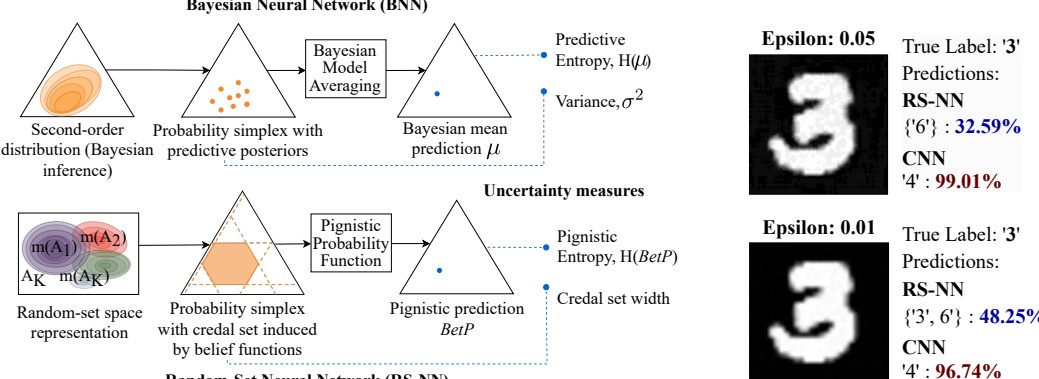

Figure 1: Inference in a Bayesian Neural Network (top) as opposed to a Random-Set Neural Network (bottom), with corresponding measures of uncertainty and their sources. The triangle represents the set of probability vectors (probability simplex) one can define on the target space (e.g., a set of 3 classes).

Figure 2: Confidence scores of RS-NN and CNN for FGSM adversarial attack for different perturbations ($\epsilon = 0.05, 0.01$) on MNIST dataset.

In a Random-Set Neural Network (Fig. 1, bottom), each test input at inference time is mapped to a belief function (Shafer, 1976) on the collection of classes $\mathcal{C}$. This belief function is encoded by a mass value $m(A) \in [0, 1]$ assigned to a finite budget $\mathcal{O} = \{A_k \subset \mathcal{C}\}$ of *sets* of classes (left). The most relevant such sets are identified from training data by fitting a Gaussian Mixture Model (GMM) to the labelled data and computing the overlap among the resulting clusters (Spruyt, 2013). Such a predicted belief function is mathematically equivalent to a convex set of probability vectors (*credal set* (Levi, 1980; Cuzzolin, 2008)) on the class list $\mathcal{C}$ (middle). A pointwise prediction (right) can then be obtained by computing the centre of mass of this credal set, termed *pignistic probability* (Smets & Kennes, 1994). In RS-NN, uncertainty can be expressed using either the entropy of the pignistic prediction (analogously to the Bayesian case), or the width of the credal set prediction (Antonucci & Cuzzolin, 2010). We find empirically that the pignistic entropy better separates in-distribution (iD) and out-of-distribution (OoD) samples than Bayesian entropy does (see Tab. 2); the width of the credal prediction, as it encodes the epistemic uncertainty about the prediction itself, is empirically much less correlated with the confidence score than entropy (see §E.5.3).

Using a random-set representation prevents the need to discard information, a challenge observed in Bayesian Model Averaging (Hinne et al., 2020; Graefe et al., 2015). While Bayesian inference requires defining prior distributions for model parameters even in the absence of relevant information, in belief theory priors are not required for the inference process, thus avoiding the selection bias risks that can seriously condition Bayesian reasoning (Freedman, 1999). Based on our extensive experiments on multiple datasets, including large-scale ones, RS-NNs not only demonstrate superior accuracy compared to state-of-the-art Bayesian and Ensemble models (Sec. 4.2), but also arguably better encode the epistemic uncertainty associated with the predictions (Sec. 4.4) and better distinguish in-distribution and out-of-distribution data (Sec. 4.3). Furthermore, RS-NNs effectively circumvent the tendency of standard networks to generate overconfident incorrect predictions, as illustrated in Fig. 2 in an experiment on Fast Gradient Sign Method (FGSM) (Goodfellow et al., 2014) adversarial attacks on MNIST (LeCun & Cortes, 2005). CNN misclassifies with high confidence scores of 99.01% and 96.74%, while RS-NN shows lower confidence at 32.59% and 48.25%.

**Contributions. Firstly,** we propose a novel *Random-Set Neural Network (RS-NN)* approach based on the principle that a deep neural network predicting belief values for *sets* of classes, rather than individual classes, has the potential to be a more faithful representation of the epistemic uncertainty induced by the limited quantity and quality of the training data. RS-NN is a 'wrapper' technique

that can be applied on top of any existing baseline network model, by just changing the output layers and loss function. Statistical guarantees can also easily be provided for RS-NN predictions by applying conformal prediction on the pignistic probabilities (see §A.4). **Secondly,** we outline a *budgeting* method for efficiently selecting a limited budget of relevant sets of classes for the task at hand given the available training data, by fitting Gaussian Mixture Models to the labelled training points and computing their clusters. This overcomes the exponential complexity of vanilla random-set implementations, ensures the scalability of the approach to large datasets, and helps the network learn by limiting the available degrees of freedom (§E.7). **Thirdly,** we introduce two new methods for assessing the uncertainty associated with a random-set prediction: the Shannon entropy of the pignistic prediction and the width of the credal prediction itself, which prove to be more robust uncertainty measures. For instance, pignistic entropy provides a clearer distinction between in-distribution (iD) and out-of-distribution (OoD) entropy (Fig. 18, §E.5.1), while credal set width excels in separating iD and OoD samples, particularly for challenging datasets like ImageNet vs. ImageNet-O (Tab. 7). **Finally,** we present a large body of experimental results (based on a fair comparison principle in which all competing models are trained from scratch) which demonstrate how RS-NN outperforms both state-of-the-art Bayesian (LB-BNN (Hobbhahn et al., 2022), FSVI (Rudner et al., 2022)) and Ensemble (DE (Lakshminarayanan et al., 2017), ENN (Osband et al., 2024)) methods in terms of: (i) performance (test accuracy, inference time) (Sec. 4.2); (ii) results on various out-of-distribution (OoD) benchmarks (Sec. 4.3), including CIFAR-10 vs. SVHN/Intel-Image, MNIST vs. FMNIST/KMNIST, and ImageNet vs. ImageNet-O; (iii) ability to provide reliable measures of uncertainty quantification (Sec. 4.4) in the form of pignistic entropy and credal set width, verified on OoD benchmarks; (iv) scalability to large-scale architectures (WideResNet-28-10, Inception V3, EfficientNet B2, ViT-Base-16) and datasets (e.g. ImageNet) (Sec. 4.5). Additionally, we show how RS-NN is robust to adversarial attacks (§E.4) and noisy data (§E.3), and circumvents the overconfidence problem in CNNs (§E.2). A qualitative assessment of entropy vs credal set width is given in §E.5.3 and in-distribution vs out-of-distribution entropy scores are shown in Fig. 18.

**Paper outline**. We first recall the notions of random sets, belief functions, credal and pignistic predictions (Sec. 2). We explain the RS-NN approach, loss function and uncertainty representation in Sec. 3. Sec. 4 provides a large body of empirical evidence supporting our approach and discusses its limitations. Sec. 5 concludes and outlines future work. Appendix §A discusses RS-NN learning process and statistical guarantees, §B reviews further related work, §C describes all algorithms in detail while §E contains a wealth of additional experimental results and all implementation details.

**Related Work.** The machine learning community has recognised the challenge of estimating uncertainty in model predictions, leading to the development of several Bayesian approximations (Gal & Ghahramani, 2016; Charpentier et al., 2020), evidential Dirichlet models (Sensoy et al., 2018; Gao et al., 2024) and conformal prediction (Shafer & Vovk, 2008; Papadopoulos et al., 2008; Balasubramanian et al., 2014; Vovk, 2012; Angelopoulos & Bates, 2021). Some methods rely on prior knowledge (Fortuin, 2022), whereas others require setting a desired threshold on predictions (Angelopoulos & Bates, 2021). Some (Baron, 1987; Hüllermeier & Waegeman, 2021) have argued that classical probability is not equipped to model 'second-level' uncertainty on the probabilities themselves. This has led to the formulation of numerous uncertainty calculi (Cuzzolin, 2020), including possibility theory (Dubois & Prade, 1990), probability intervals (Halpern, 2017), credal sets (Levi, 1980), random sets (Nguyen, 1978) or imprecise probability (Walley, 1991).

Bayesian approaches, pioneered by Buntine & Weigend (1991) and others (MacKay, 1992; Neal, 2012), are dominant in uncertainty estimation. Notable techniques include R-BNN (Reparameterisation) (Kingma et al., 2015), variational inference with reparameterisation and Laplace Bridge Bayesian approximation (LB-BNN) (Hobbhahn et al., 2022), which uses the Laplace Bridge to map between Gaussian and Dirichlet distributions. Various approximations of full Bayesian inference exist, such as Markov chain Monte Carlo (MCMC), function-space BNNs (Sun et al., 2019) such as function-space variational inference (FSVI) (Rudner et al., 2022), and Dropout Variational Inference (Gal & Ghahramani, 2015). In our experiments, we do not consider older Bayesian models such as R-BNN, MCMC and Dropout VI, since they have been superseded by more performing approaches and are computationally expensive to train on larger datasets and architectures. Recent work addresses challenges in computational cost (Hobbhahn et al., 2022; Daxberger et al., 2021) and model priors (Tran et al., 2020). Despite their advantages, Bayesian models face challenges when the model prior is misspecified. Ensemble-based approaches, such as Deep Ensembles (DE) (Lakshminarayanan et al., 2017) and Epistemic Neural Networks (ENN) (Osband et al., 2024), efficiently estimate uncertainty by leveraging multiple models. However, the computational cost of

training ensembles, especially for large models, is often impractical. Our approach mitigates these challenges by eliminating the need for both inference-time sampling and prior selection, reducing computational complexity compared to Bayesian inference and lowering training time compared to Ensembles, as demonstrated in our experiments in Sec. 4. More related work is given in §B.

## 2 RANDOM SETS AND BELIEF FUNCTIONS

**Random sets.** A die is a simple example of a (discrete) random variable. Its probability space is defined on the sample space $\Theta = \{\text{face}1, \text{face }2, \ldots, \text{face } 6\}$, where elements are mapped to the real numbers $1, 2, \ldots, 6$, respectively. Now, imagine that faces 1 and 2 are cloaked, and we roll the die. How do we model this new experiment, mathematically? Actually, the probability space has not changed (as the physical die has not been altered, its faces still have the same probabilities). What has changed is the mapping: since we cannot observe the outcome when a cloaked face is shown (assuming that only the top face is observable), both face 1 and face 2 (as elements of $\Theta$) are mapped to the set of possible values $\{1, 2\}$ on the real line $\mathbb{R}$ (Fig. 3). Mathematically, this is a *random set* (Matheron, 1975; Kendall, 1974; Nguyen, 1978; Molchanov, 2005), i.e., a set-valued random variable, modelling random experiments in which observations come in the form of sets.

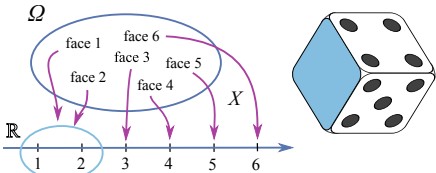

Figure 3: The random set associated with a cloaked die in which faces 1 and 2 are not visible.

Figure 4: A belief function is equivalent to a credal set with boundaries determined by lower bounds (Eq. 4) on probability values.

**Belief functions.** Random sets have been proposed by Dempster (2008) and Shafer (1976) as a mathematical model for subjective belief, alternative to Bayesian reasoning. Thus, on finite domains (e.g., for classification) they assume the name of *belief functions*. While classical discrete mass functions assign normalised, non-negative values to *elements* $\theta \in \Theta$ of their sample space, a belief function independently assigns normalised, non-negative mass values to its *subsets* $A \subset \Theta$:

$$m(A) \geq 0, \quad \sum_A m(A) = 1, \ \forall A \in \mathbb{P}(\Theta) \tag{1}$$

The belief function associated with a mass function $m$ measures the total mass of the subsets of each 'focal set' $A$. Mass functions can be recovered from belief functions via Moebius inversion (Shafer, 1976), which, in combinatorics, plays a role similar to that of the derivative:

$$Bel(A) = \sum_{B \subseteq A} m(B), \quad m(A) = \sum_{B \subseteq A} (-1)^{|A \setminus B|} Bel(B). \tag{2}$$

**Example.** Consider a sample space (class list) $\Theta = \mathcal{C} = \{c_1, c_2, c_3\}$ and let $\mathbb{P}(\Theta)$ be its power set (collection of all subsets). As shown in Fig. 4, one can define a mass function on $\mathbb{P}(\Theta)$ as: $m(\{c_1\}) = 0.5$, $m(\{c_3\}) = 0.1$, $m(\{c_1, c_2\}) = 0.4$ (Fig. 4, top), and the masses for all other sets (unspecified) equal to zero. Note that $m$ is normalised: $\sum_{B \subseteq \Theta} m(B) = 1$. By Eq. 2, the belief value of the composite class $A = \{c_2, c_3\}$ is: $Bel(\{c_2, c_3\}) = m(\{c_3\}) = 0.1$ (Fig. 4, left). Similarly, the belief value of composite class $\{c_1, c_2\}$ can by computed as $Bel(\{c_1, c_2\}) = m(\{c_1\}) + m(\{c_1, c_2\}) = 0.5 + 0.4 = 0.9$.

**Pignistic probability.** Given a belief function $Bel$, its *pignistic probability* is the precise probability distribution obtained by re-distributing the mass of its focal sets $A$ to its constituent elements, $\theta \in A$:

$$BetP(\theta) = \sum_{A \ni \theta} \frac{m(A)}{|A|}. \tag{3}$$

Smets (2005) originally proposed to use the pignistic probability for decision making using belief functions, by applying expected utility to it. Notably, the pignistic probability is geometrically the centre of mass of the credal set (see Fig. 4) associated with a belief function (Cuzzolin, 2018).

**Credal prediction.** As anticipated, RS-NN is designed to predict a belief function (a finite random set) on the set of classes. A belief function, in turn, is associated with a convex set of probability distributions (a *credal set* (Levi, 1980; Zaffalon & Fagiuoli, 2003; Cuzzolin, 2010; Antonucci & Cuzzolin, 2010; Cuzzolin, 2008)) on the same domain. This is the set:

$$Cre = \{P : \Theta \to [0, 1] | Bel(A) \leq P(A)\}, \tag{4}$$

of probability distributions $P$ on $\Theta$ which dominate the belief function on each focal set $A$. The size of the resulting credal prediction measures the extent of the related epistemic uncertainty (see Sec. 3.3), arising from lack of evidence (see §E.5.3). The use of credal set size as a measure of epistemic uncertainty is well-supported in literature (Hüllermeier & Waegeman, 2021; Bronevich & Klir, 2008), as it aligns with established concepts of uncertainty such as conflict and non-specificity (Yager, 2008; Kolmogorov, 1965). A credal prediction, by encompassing multiple potential distributions, reflects the model's acknowledgment of this uncertainty. A wider credal set indicates higher uncertainty, as the model refrains from committing to a specific probability distribution due to limited or conflicting evidence. In contrast, a narrower credal set implies lower uncertainty, signifying a more confident prediction based on substantial, consistent evidence.

# 3 RANDOM-SET NEURAL NETWORK

## 3.1 APPROACH

**Representation.** As shown in Fig. 5 (b), RS-NN predicts for each input data point a belief function, rather than a vector of softmax probabilities as in a traditional CNN. For $N$ classes, a 'vanilla' RS-NN would have $2^N$ outputs (as $2^N$ is the cardinality of $\mathbb{P}(\mathcal{C})$), each being the belief value (Eq. 2) of the focal set of classes $A \in \mathbb{P}(\mathcal{C})$ corresponding to that output neuron. Our architecture focuses primarily on the final layers (Fig. 5 (b), in grey), acting as a wrapper applicable on top of *any* baseline representation layers from existing models (Fig. 5 (b), in blue). This enables the integration of pre-trained networks while fine-tuning only the final decision layers. Hence, RS-NN is easily scalable to any model architecture, as demonstrated in Sec. 4.5, Tab. 4.

Given a training datapoint with a true class attached, its ground truth is encoded by the vector $\mathbf{bel} = \{Bel(A), A \in \mathbb{P}(\mathcal{C})\}$ of belief values for each focal set of classes $A \in \mathbb{P}(\mathcal{C})$. $Bel(A)$ is set to 1 iff the true class is contained in the subset $A$, 0 otherwise[1]. This corresponds to full certainty that the element belongs to that set and there is complete confidence in this proposition (see §A.2 for an example).

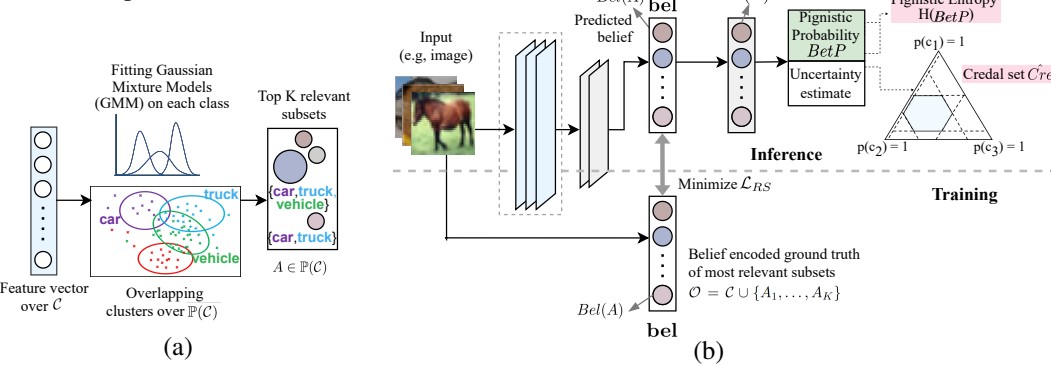

Figure 5: RS-NN model architecture. (a) *Budgeting*: Given a collection $\mathcal{C}$ of $N$ classes, the top $K$ relevant (focal) sets of classes $\{A_1, \ldots, A_K\}$ are selected from the powerset $\mathbb{P}(\mathcal{C})$ (Algorithm in §C.1) and added to the singleton classes to form the budget $\mathcal{O}$. (b) *Training and Inference*: Ground truth classes are encoded as belief vectors $\mathbf{bel}$ and used to predict a belief function $\hat{\mathbf{bel}}$ for each training data point by minimising the loss $\mathcal{L}_{RS}$ (7), to train the output layers (in grey) producing $\hat{\mathbf{bel}}$. Mass values $\hat{m}$ and pignistic probability estimates $BetP$ are computed from the predicted belief function. Uncertainty is estimated as described in Sec. 3.3.

**Budgeting**. To overcome the exponential complexity of using $2^N$ sets of classes (especially for large $N$), a fixed budget of $K$ relevant non-singleton (of cardinality $> 1$) focal sets are used. These focal sets are obtained by clustering the original classes $\mathcal{C}$, fitting ellipses over them, and selecting the top $K$ focal sets of classes with the highest overlap ratio (Fig. 5 (a)). This is computed as the intersection over union for each subset $A$ in $\mathbb{P}(\mathcal{C})$: $overlap(A) = \cap_{c \in A} A^c / \cup_{c \in A} A^c$, $A_1, \ldots, A_K \in \mathbb{P}(\mathcal{C})$. Clustering is performed on feature vectors of images generated by any feature extractor trained on the original classes $\mathcal{C}$. In our experiments, we used features from a trained standard ResNet50 model. These feature vectors are further reduced to 3 dimensions using t-SNE (t-Distributed Stochastic Neighbor Embedding) (van der Maaten & Hinton, 2008) before applying a Gaussian Mixture Model

---

[1]Note that the belief encoding of ground-truth is not related to label smoothing. It maps ground-truth labels from the original class space to a set space without adding noise, thus preserving label "hardness".

(GMM) to them. Ellipsoids (Spruyt, 2013), covering $95\%$ of data, are generated using eigenvectors and eigenvalues of the covariance matrix $\Sigma_c$ and the mean vector $\mu_c$, $\forall c \in \mathcal{C}$, $P_c \sim \mathcal{N}(x_c; \mu_c, \Sigma_c)$ obtained from the GMM to calculate the overlaps. To avoid computing a degree of overlap for all $2^N$ subsets, the algorithm is early stopped when increasing the cardinality does not alter the list of most-overlapping sets of classes. The $K$ non-singleton focal sets so obtained, along with the $N$ original (singleton) classes, form our network's budget of outputs $\mathcal{O} = \mathcal{C} \cup \{A_1, \ldots, A_K\}$. E.g., in a 100-class scenario, the powerset contains $2^{100}$ subsets ($10^{30}$ possibilities). Setting a budget of $K = 200$, for instance, results in $100 + K = 300$ outputs, a far more manageable number.

As shown in our experiments, budgeting also helps the network converge without overfitting. While, in theory, a complete belief function model would be more powerful, in practice a larger number of focal sets impedes the learning process. As shown in Tab. 16, §E.7, RS-NN with a limited number of well-representative sets often performs better than RS-NN with a full power set of classes, and is more efficient at uncertainty estimation. The budgeting step is a one-time procedure, applied to any given dataset before training. It requires just 2 minutes for CIFAR-10, 7 minutes for CIFAR-100, and 60 minutes for ImageNet ($\approx 1.1$M images, 1000 classes).

Further, our budgeting procedure is not specific to t-SNE, but can use any dimensionality reduction technique (Maćkiewicz & Ratajczak, 1993). For instance, Uniform Manifold Approximation and Projection (UMAP) (McInnes et al., 2018) requires approximately 1 minute for the CIFAR-10 dataset, 2 minutes for CIFAR-100, and around 23 minutes for ImageNet to generate embeddings (see Tabs. 17, 18 in §E.8 for a t-SNE vs UMAP ablation study). This is including the overlap computation which only takes a few seconds and is efficiently parallelised across 150 CPU cores.

## 3.2 LOSS FUNCTION

A random-set prediction problem is mathematically similar to the multi-label classification problem, for in both cases the ground truth vector contains several 1s. In the former case, these correspond to sets all containing the true class; in the latter, to all the class labels attached to same data point. Despite the different semantics, we can thus adopt as loss Binary Cross-Entropy (BCE) (5) with sigmoid activation, to drive the prediction of a belief value for each focal set in the identified budget:

$$\mathcal{L}_{BCE} = -\frac{1}{b_{size}} \sum_{i=1}^{b_{size}} \frac{1}{|\mathcal{O}|} \sum_{A \in \mathcal{O}} \Big[ Bel_i(A) \log(\hat{Bel}_i(A)) + (1 - Bel_i(A)) \log(1 - \hat{Bel}_i(A)) \Big]. \tag{5}$$

Here, $i$ is the index of the training point within a batch of cardinality $b_{size}$, $A$ is a focal set of classes in the budget $\mathcal{O}$, $Bel_i(A)$ is the $A$-th component of the vector $\mathbf{bel}_i$ encoding the ground truth belief values for the $i$-th training point, and $\hat{Bel}_i(A)$ is the corresponding belief value in the predicted vector $\hat{\mathbf{bel}}_i$ for the same training point. Both $\mathbf{bel}_i$ and $\hat{\mathbf{bel}}_i$ are vectors of cardinality $|\mathcal{O}|$ for all $i$. A valid belief function satisfies Eq. 1 which states that mass values derived from belief functions should be non-negative and should sum up to 1 (Shafer, 1976). To ensure this, we incorporate a mass regularization term $M_r$ and a mass sum term $M_s$ in the loss function:

$$M_r = \frac{1}{b_{size}} \sum_{i=1}^{b_{size}} \sum_{A \in \mathcal{O}} \max(0, -\hat{m}_i(A)), \quad M_s = \max\left(0, \frac{1}{b_{size}} \sum_{i=1}^{b_{size}} \sum_{A \in \mathcal{O}} \hat{m}_i(A) - 1\right). \tag{6}$$

$M_r$ encourages non-negativity of the (predicted) mass values $\hat{m}(A)$, $A \in \mathcal{O}$. These mass values are obtained from the predicted belief function $\hat{\mathbf{bel}}$ via the Moebius inversion formula (Eq. 2). For it to be valid, the sum of the masses of the predicted belief function must be equal to 1 (Eq. 1), which is encouraged by the mass sum term $M_s$. All loss components $\mathcal{L}_{BCE}$, $M_r$ and $M_s$ are computed during batch training. Two hyperparameters, $\alpha$ and $\beta$, control the relative importance of the two mass terms, yielding as the total loss for RS-NN:

$$\mathcal{L}_{RS} = \mathcal{L}_{BCE} + \alpha M_r + \beta M_s. \tag{7}$$

The regularisation terms aim to penalise deviations from valid belief functions, in line with, e.g., the way training time regularisation in neurosymbolic learning encourages predictions to be common-sense (Giunchiglia et al., 2023). In general, soft constraints (Márquez-Neila et al., 2017) have been shown to not be inferior to hard ones. Still, when $\alpha$ and $\beta$ are too small, this may not be sufficient to ensure that predictions are valid belief functions. In such cases,[2] post-training, we set any negative

---

[2] At any rate, improper belief functions are normally used in the literature (Denoeux, 2021), e.g. for conditioning (Cuzzolin, 2020).

masses to zero and add the 'universal' set of all classes to the final budget. This subset is assigned all the remaining mass, ensuring that the sum of masses across all focal sets in $\mathcal{O}$ equals 1. This approach mimics classical approximation schemes (e.g. Cuzzolin (2020), Part III).

## 3.3 Accuracy and uncertainty estimation

**Pignistic prediction**. The pignistic probability (Eq. 3) is the central prediction associated with a belief function (seen as a credal set): standard performance metrics such as accuracy can then be calculated after extracting the most likely class according to the pignistic prediction (e.g. in §E.5.1). Notably, the RS-NN architecture and training mechanism are designed to facilitate set-based learning by incorporating class set information during training. When pignistic probabilities are computed during inference, they reflect the learning derived from the masses and beliefs of various subsets, making it more reliable than traditional softmax probabilities, as shown in §E.2.

**Entropy of the pignistic prediction**. The Shannon entropy of the pignistic prediction $BetP$ can then be used as a measure of the uncertainty associated with the predictions, in some way analogous to the entropy of a Bayesian mean average prediction:

$$H_{RS} = -\sum_{c \in \mathcal{C}} BetP(c) \log BetP(c). \tag{8}$$

A higher entropy value (8) indicates greater uncertainty in the model's predictions.

**Size of the credal prediction**. As discussed above, and further elaborated upon in §E.5.3, a sensible measure of the epistemic uncertainty attached to a random-set prediction $\hat{bel}$ is the size of the corresponding credal set. Several ways exist of measuring the size of a convex polytope such as a credal set (Sale et al., 2023). Given the upper and lower bounds to the probability assigned to each class $c$ by distributions within the predicted credal set $\hat{Cre}$ (Eq. 4),

$$\overline{P}(c) = \max_{P \in Cre} P(c), \quad \underline{P}(c) = \min_{P \in Cre} P(c), \tag{9}$$

we propose a simple way to measure the size of the credal set as the difference between the lower and upper bounds (Eq. 9) associated with the most likely class, according to the pignistic prediction. Note that the predicted pignistic estimate $BetP(c)$ falls within the interval $[\underline{P}(c), \overline{P}(c)]$ for each class $c$, not just for the most likely class $\hat{c}$. Its width $\overline{P}(c) - \underline{P}(c)$ indicates the epistemic uncertainty associated with the prediction (§A.3).

## 4 Experiments

### 4.1 Implementation

**Datasets.** Our experiments are performed on multi-class image classification datasets, including MNIST (LeCun & Cortes, 2005), CIFAR-10 (Krizhevsky et al., 2009), Intel Image (Bansal, 2019), CIFAR-100 (Krizhevsky, 2012), and ImageNet (Deng et al., 2009). For out-of-distribution (OoD) experiments, we assess several in-distribution (iD) vs OoD datasets: CIFAR-10 vs SVHN (Netzer et al., 2011)/Intel-Image (Bansal, 2019), MNIST vs F-MNIST (Xiao et al., 2017)/K-MNIST (Clanuwat et al., 2018), and ImageNet vs ImageNet-O (Hendrycks et al., 2021). The data is split into 40000:10000:10000 samples for training, testing, and validation respectively for CIFAR-10 and CIFAR-100, 50000:10000:10000 samples for MNIST, 13934:3000:100 for Intel Image, 1172498:50000:108669 for ImageNet. For OoD datasets, we use 10,000 testing samples, except for Intel Image (3,000) and ImageNet-O (2,000). Training images are resized to $224 \times 224$ pixels.

**Baselines, backbone and training details.** Our baselines include state-of-the-art Bayesian methods LB-BNN (Hobbhahn et al., 2022) and FSVI (Rudner et al., 2022), Ensemble classifiers DE (5 ensembles) (Lakshminarayanan et al., 2017) and ENN (3 ensembles) (Osband et al., 2024), and standard CNN (see Sec. 4.2. Our baseline models are similar to those used in the most recent work (Cinquin & Bamler, 2024; Daxberger et al., 2021; Wu & Williamson, 2024) in uncertainty estimation. All models, including RS-NN, are trained on ResNet50 (on NVIDIA A100 80GB GPUs) with a learning rate scheduler initialized at 1e-3 with 0.1 decrease at epochs 80, 120, 160 and 180. Standard data augmentation (Krizhevsky et al., 2012), including random horizontal/vertical shifts with a magnitude of 0.1 and horizontal flips, is applied to all models.

ResNet50, excluding the top classification layer, serves as the common architecture. ResNet was originally designed for classification on the ImageNet dataset (1000 classes). To accommodate a reduced number of classes as in smaller datasets (e.g., CIFAR-10/MNIST with 10 classes, Intel Image with 7 classes, and CIFAR-100 with 100 classes), two additional dense layers (1024 and 512

Table 1: Test accuracies (%) and inference time (ms) for uncertainty estimation over 5 consecutive runs across methods and datasets. Average and standard deviation are shown for each experiment.

| Datasets | MNIST | CIFAR-10 | Intel Image | CIFAR-100 | ImageNet (Top-1) | ImageNet (Top-5) | Inference time (ms) |
|---|---|---|---|---|---|---|---|
| **RS-NN** (ours) | **99.71 ± 0.03** | **93.53 ± 0.09** | **94.22 ± 0.03** | **71.61 ± 0.07** | **79.92** | **94.47** | **1.91 ± 0.02** |
| LB-BNN (Hobbhahn et al., 2022) | 99.58 ± 0.04 | 89.95 ± 0.81 | 90.49 ± 0.42 | 59.89 ± 1.96 | 72.48 | 90.85 | 7.11 ± 0.89 |
| FSVI (Rudner et al., 2022) | 99.18 ± 0.03 | 80.29 ± 0.05 | 88.92 ± 0.13 | 53.34 ± 0.09 | 62.56 | 84.69 | 340.25 ± 0.76 |
| DE (Lakshminarayanan et al., 2017) | 99.25 ± 0.01 | 92.73 ± 0.04 | 91.98 ± 0.11 | 70.53 ± 0.07 | 78.77 | 94.37 | 13163.50 ± 3.37 |
| ENN (Osband et al., 2024) | 99.07 ± 0.11 | 91.55 ± 0.60 | 91.49 ± 0.19 | 68.02 ± 0.26 | 71.82 | 89.48 | 3.10 ± 0.03 |
| CNN | 99.12 ± 0.04 | 92.08 ± 0.42 | 90.89 ± 0.10 | 65.50 ± 0.08 | 78.56 | 94.34 | 1.91 ± 0.03 |

neurons, ReLU activation) are added to ResNet50. Similar techniques are commonly applied in deep learning to adapt model architectures to datasets (He et al., 2016; Zagoruyko & Komodakis, 2016). The output layer of RS-NN on ResNet50 has the same number of units as the number of (selected) focal sets $|\mathcal{O}|$, and uses sigmoid activation, since the ground truth encoding resembles a multi-label classification problem (see Sec. 3.2). For all other models, the final output layer simply consists of a softmax activation for multi-class classification. All the models were trained from scratch for 200 epochs (recommended by most), with a batch size of 128. *Our objective is to ensure a fair comparison across all models for all experiments.* Tuning each model separately to maximise performance would not guarantee that, as some models use pre-trained weights while others train for larger number of epochs, which could result in the over-training of another model. We used pre-trained weights for all models only when trained on ImageNet for efficiency. For all other training hyperparameters (e.g, optimizer), we use what is specified for each model in their respective papers (more in §D, Tab. 5). Consequently, the performance metrics reported in the original papers may vary from those presented here, as they were obtained under different training conditions.

**Training the RS-NN.** In the clustering phase, we obtain features directly from a pre-trained ResNet50 classifier. We use 50 CPU cores for t-SNE dimensionality reduction, and 150 CPU cores for computing the overlap of classes. We set a budget $K$ of 20 focal sets (ablation study on K in §E.7) for CIFAR-10/ MNIST/ Intel Image, 200 for CIFAR-100 and 3000 for ImageNet. RS-NN is trained from scratch on ground-truth belief encoding of sets using the $\mathcal{L}_{RS}$ loss function (Eq. 7) over 200 epochs, with a batch size ($b_{size}$) of 128 and $\alpha = \beta = 1e-3$ as hyperparameter values.

**Experiments. Firstly**, we assess the *test accuracy* (%) and *inference time* (ms) of all baselines on all the multi-class classification datasets. Tab. 1 provides a comparison of test accuracies and inference times of RS-NN against the baselines. Our **second** set of experiments (Sec. 4.3) concerns *out-of-distribution (OoD) detection*. Tab. 2 shows our results on OoD detection metrics AUROC (Area Under Receiver Operating Characteristic curve) and AUPRC (Area Under Precision-Recall curve) for all the models on the iD vs OoD datasets listed above. We also report the Expected Calibration Error (ECE) for all models on datasets CIFAR-10, MNIST and ImageNet (see Tab. 2). In a **third** set of experiments (Sec. 4.4), we test the *uncertainty estimation* capabilities of RS-NN using both the entropy of the pignistic prediction and the size of the predicted credal set. Tab. 2 presents the predicted pignistic entropy of RS-NN alongside entropies of all baselines on iD and OoD datasets. Tab. 3 shows credal set widths for RS-NN predictions across the same datasets. In our **final** set of experiments, we explore the *scalability of RS-NN* (Sec. 4.5) to large-scale architectures, employing it on models such as WideResNet-28-10, VGG16, Inception V3, EfficientNetB2 and ViT-Base-16. Further, to underscore the model's ability to leverage transfer learning, we train and test on a pre-trained ResNet50 model initialised with ImageNet weights (Tab. 4).

**Additional experiments** concern obtaining statistical guarantees for RS-NN using non-conformity scores (§A), evaluating the robustness of RS-NN to adversarial attacks (§E.4), noisy and rotated in-distribution samples (Appendix §E.3), showing how RS-NN circumvents the overconfidence problem in CNNs (§E.2). Results showing how credal set width is less correlated with confidence scores than entropy are discussed in in §E.5.3. We also conduct ablation studies on $\alpha$ and $\beta$ (§E.6) and the number of non-singleton focal sets $K$ (§E.7), which show that the best accuracy is obtained at small values ($1e-3$) of $\alpha$ and $\beta$, and for $K = 20$ (on the CIFAR-10 dataset).

## 4.2 COMPARISON WITH THE STATE-OF-THE-ART ON ACCURACY

Tab. 1 shows that RS-NN outperforms state-of-the-art Bayesian (LB-BNN, FSVI) and Ensemble (DE, ENN) methods in terms of test accuracy (%) across all datasets. Experiments on inference time (ms) per sample, conducted over 5 runs (a single forward propagation) on the CIFAR-10 dataset, show that RS-NN and CNN have the fastest inference times among all models (Tab. 1). Albeit CNN and DE are close to RS-NN in performance (e.g., on ImageNet), CNN tends to be overconfident in incorrect predictions (§E.2), while DE has significantly longer training (Tab. 6, §D) and inference times (Tab. 1). The pignistic predictions derived from predicted belief functions are more reliable and accurate than predicted softmax probabilities of CNN and DE. It is important to

Table 2: OoD detection performance and uncertainty estimation for models trained on ResNet50 on CIFAR-10 vs SVHN/Intel Image, MNIST vs F-MNIST/K-MNIST and ImageNet vs ImageNet-O. Evaluation metrics include AUROC/AUPRC (OoD); Entropy of predictions (uncertainty) and Expected Calibration Error (ECE).

| | | In-distribution (iD) | | | | Out-of-distribution (OoD) | | | | | |
| | | | | | | SVHN | | | Intel Image | | |
| Dataset | Model | Test accuracy (%) (↑) | Uncertainty measure | In-distribution Entropy (↓) | ECE (↓) | AUROC (↑) | AUPRC (↑) | Entropy (↑) | AUROC (↑) | AUPRC (↑) | Entropy (↑) |
|---|---|---|---|---|---|---|---|---|---|---|---|
| CIFAR-10 | RS-NN | **93.53** | Pignistic entropy | **0.088 ± 0.308** | 0.0484 | **94.91** | **93.72** | **1.132 ± 0.855** | **97.39** | **90.27** | **1.517 ± 0.740** |
| | LB-BNN | 89.95 | Predictive Entropy | 0.191 ± 0.412 | 0.0585 | 88.14 | 81.96 | 0.828 ± 0.243 | 82.21 | 55.17 | 0.763 ± 0.722 |
| | FSVI | 80.29 | Predictive Entropy | 0.118 ± 0.563 | 0.0521 | 80.59 | 80.84 | 0.413 ± 0.461 | 74.27 | 72.51 | 0.289 ± 0.670 |
| | DE | 92.73 | Mean Entropy | 0.154 ± 0.367 | **0.0482** | 93.84 | 91.88 | 0.939 ± 0.554 | 94.25 | 79.36 | 1.166 ± 0.552 |
| | ENN | 91.55 | Mean Entropy | 0.126 ± 0.323 | 0.0556 | 92.76 | 89.05 | 0.887 ± 0.514 | 85.67 | 58.09 | 0.600 ± 0.578 |
| | CNN | 92.08 | Softmax Entropy | 0.114 ± 0.304 | 0.0669 | 93.11 | 91.0 | 0.930 ± 0.610 | 87.75 | 65.54 | 0.719 ± 0.673 |
| | | | | | | F-MNIST | | | K-MNIST | | |
| MNIST | RS-NN | **99.71** | Pignistic entropy | 0.010 ± 0.111 | **0.0029** | **93.89** | **93.98** | 0.530 ± 0.770 | **96.75** | **96.58** | **0.740 ± 0.917** |
| | LB-BNN | 99.58 | Predictive Entropy | **0.001 ± 0.085** | 0.0032 | 89.65 | 90.36 | 0.287 ± 0.442 | 95.61 | 95.65 | 0.540 ± 0.621 |
| | FSVI | 99.18 | Predictive Entropy | 0.006 ± 0.265 | 0.0047 | 92.79 | 91.17 | 0.264 ± 0.289 | 95.75 | 95.75 | 0.313 ± 0.381 |
| | DE | 99.25 | Mean Entropy | 0.031 ± 0.155 | 0.0031 | 92.30 | 92.05 | **0.584 ± 0.587** | 95.81 | 94.71 | 0.564 ± 0.715 |
| | ENN | 99.07 | Mean Entropy | 0.022 ± 0.127 | 0.0039 | 81.79 | 82.92 | 0.313 ± 0.464 | 95.94 | 95.45 | 0.503 ± 0.672 |
| | CNN | 98.90 | Softmax Entropy | 0.023 ± 0.135 | 0.0052 | 83.77 | 84.14 | 0.278 ± 0.426 | 94.46 | 93.94 | 0.616 ± 0.688 |
| | | | | | | ImageNet-O | | | | | |
| | | | | | | AUROC | | AUPRC | | Entropy | |
| ImageNet | RS-NN | **79.92** | Pignistic entropy | 2.972 ± 2.108 | 0.1416 | **60.38** | | **55.16** | | 3.659 ± 3.771 | |
| | LB-BNN | 72.48 | Predictive Entropy | 2.471 ± 2.972 | 0.5812 | 41.08 | | 30.99 | | 1.383 ± 0.028 | |
| | FSVI | 62.56 | Predictive Entropy | **1.328 ± 1.966** | 0.3890 | 50.55 | | 49.88 | | 1.637 ± 1.328 | |
| | DE | 78.77 | Mean Entropy | 1.532 ± 1.325 | 0.1940 | 55.37 | | 53.20 | | 1.775 ± 1.343 | |
| | ENN | 71.82 | Mean Entropy | 1.395 ± 1.510 | 0.5961 | 54.67 | | 43.73 | | 1.617 ± 1.597 | |
| | CNN | 78.56 | Softmax Entropy | 6.386 ± 1.388 | 0.4004 | 54.28 | | 48.73 | | **6.575 ± 1.512** | |

note that the comparison with standard CNN is not based on its role as an uncertainty method, but as a benchmark, underscoring RS-NN's effectiveness in achieving high accuracy compared to other uncertainty baselines. Test accuracy for RS-NN is calculated by determining the class with the highest pignistic probability for each prediction and comparing it with the true class. This proves that, by modelling the epistemic uncertainty about the prediction, we take better into account possible distribution shifts at test time - as a result, the central prediction of the set of probabilities (credal set) associated with the predicted belief function is more likely to be closer to the ground truth. Note that the results in the FSVI paper (Rudner et al., 2022) are based on ResNet18. Although using a pre-trained ResNet50 could enhance FSVI's performance, it would provide an unfair advantage over the other baseline models in our paper, which were all trained from scratch.

## 4.3 OUT-OF-DISTRIBUTION (OoD) DETECTION

We evaluate our out-of-distribution (OoD) detection (Tab. 2) against baselines using AUROC and AUPRC scores. OoD detection identifies data points deviating from the in-distribution (iD) training data by measuring true and false positive rates (AUROC) and precision and recall trade-offs (AUPRC), indicating the model's ability to handle unfamiliar data (further detailed in §C.3). As shown in Tab. 2, RS-NN greatly outperforms Bayesian, Ensemble and standard CNN models in OoD detection, with significantly higher AUROC and AUPRC scores for all iD vs OoD datasets, especially on difficult datasets like ImageNet and ImageNet-O (see §E.1). Even high-accuracy models like CNN and DE struggle to differentiate between ImageNet and ImageNet-O. While DE shows comparable AUROC scores on CIFAR-10 vs. SVHN and MNIST vs. F-MNIST/K-MNIST to RS-NN, it has lower AUPRC. While AUROC assesses a model's ability to distinguish between iD and OoD samples, AUPRC measures both the precision of correctly identified OoD samples and the recall of detected OoD samples, making it a more informative metric. Tab. 2 also reports the various models' Expected Calibration Error (ECE) (detailed in §C.2). A low ECE indicates that the model's confidence scores align closely with the actual likelihood of events. RS-NN exhibits the lowest ECE indicating well-calibrated probabilistic predictions.

## 4.4 UNCERTAINTY ESTIMATION

**Entropy.** Tab. 2 shows the mean and variance of the entropy distributions for both iD and OoD datasets. Lower entropy values for iD datasets indicate a confident prediction as the model is trained on familiar data, while higher entropy for OoD datasets reflects the model's uncertainty with unseen data. RS-NN exhibits both low entropy for iD datasets and high entropy for OoD ones. The percentage mean iD/OoD shift in entropy is most pronounced in RS-NN, especially evident in CIFAR-10 vs SVHN/Intel Image and MNIST vs F-MNIST/K-MNIST. For ImageNet vs ImageNet-O, CNN exhibits high entropy for both iD and OoD datasets while RS-NN maintains a desirable iD vs OoD entropy ratio. In fact, Fig. 18, §E.5.1 show that RS-NN has the best iD vs OoD entropy ratio for all datasets. This ideal behavior, along with superior OoD detection (Sec. 4.3), is a good indication that RS-NN is better at estimating uncertainty induced by domain shift (see §E.5.1).

**Credal set width.** Fig. 6 shows a density plot of estimated credal set widths, the difference between the lower and upper probability bounds in Eq. 9, for correctly and incorrectly classified samples of

Table 3: Credal set width for RS-NN on iD vs OoD datasets: CIFAR10 vs SVHN/Intel Image, MNIST vs F-MNIST/K-MNIST and ImageNet vs ImageNet-O.

| In-distribution (iD) | | Out-of-distribution (OoD) | | | |
|---|---|---|---|---|---|
| CIFAR10 | $0.007 \pm 0.044$ | SVHN | $0.260 \pm 0.322$ | Intel Image | $0.587 \pm 0.367$ |
| MNIST | $0.001 \pm 0.013$ | F-MNIST | $0.070 \pm 0.167$ | K-MNIST | $0.103 \pm 0.200$ |
| ImageNet | $0.238 \pm 0.266$ | ImageNet-O | | $0.272 \pm 0.275$ | |

Table 4: Adaptability to large-scale model architectures with test accuracy (%) and parameters (in million) reported on CIFAR10.

| | Model | Pre-trained R50 | WRN-28-10 | VGG16 | IncepV3 | ENetB2 | ViT-Base |
|---|---|---|---|---|---|---|---|
| Test acc. (%) | RS-NN | **94.42** | **93.58** | **87.87** | 78.24 | **92.10** | 86.75 |
| | CNN | 94.38 | 92.79 | 84.14 | 76.89 | 90.02 | **87.21** |
| Params (M) | RS-NN | 2.69 | 37.0 | 15.12 | 31.22 | 7.72 | 9.53 |
| | CNN | 2.62 | 36.7 | 15.11 | 31.21 | 7.71 | 9.52 |

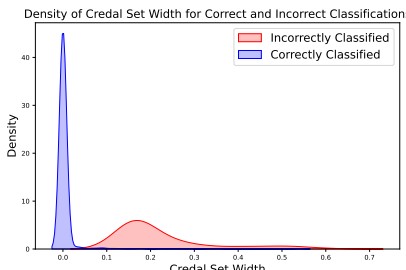

Figure 6: Width of credal predictions for CIFAR-10 test data (correctly classified, blue; incorrectly classified, red).

CIFAR-10 test data. Incorrect predictions (red) correlate with larger credal set widths, indicating higher epistemic uncertainty, and are more dispersed in the plot. Correct predictions (blue) concentrate around smaller values, exhibiting smaller credal intervals as expected. Tab. 3 reports the credal set widths of RS-NN predictions for iD vs OoD datasets. Larger intervals can be observed for OoD datasets. The credal set widths for ImageNet vs ImageNet-O are not as distinct given the nature of these datasets, as detailed in §E.1. Note that credal set width is not directly comparable to the entropy, variance or mutual information of other models, as such metrics have distinct semantics.

### 4.5 SCALABILITY TO LARGE-SCALE ARCHITECTURES

Tab. 4 shows the scalability of RS-NN to larger model architectures and its ability to leverage transfer learning. RS-NN outperforms standard CNNs across various large-scale model architectures, including WideResNet28-10 (WRN-28-10), VGG16, InceptionV3 (IncV3), EfficientNetB2 (ENetB2), and Vision Transformer (ViT-Base-16), highlighting its versatility and ease in adopting different architectures, and the generality of the random-set concept. Notably, using a pre-trained ResNet50 (Pre-trained R50) model with ImageNet weights, RS-NN achieves higher accuracy on CIFAR-10 compared to using only the architecture (no pre-trained weights).

### 4.6 LIMITATIONS

The budgeting step can be time consuming but, for a given dataset, it is a one-time procedure prior to training. For large datasets, t-SNE and GMM can be applied to representative samples rather than the entire dataset, as the purpose is merely to identify the most relevant sets, dramatically reducing computational demands. To further optimise the method, instead of t-SNE, budgeting can utilize various methods such as autoencoders (Ghasedi Dizaji et al., 2017; Guo et al., 2017b) or PCA (Abdi & Williams, 2010) for dimensionality reduction. However, despite the initial time investment, the efficiency gained from having a budget applicable to any training scenario is substantial. Another limitation is manually setting the number of focal elements $K$; a dynamic strategy adjusting $K$ based on overlap would enhance flexibility and effectiveness. The budgeting procedure can also be adapted to work with multiple sources of data or a stream of data (§E.8). Alternative methods such as using sparse mass functions (Itkina et al., 2020; Chen et al., 2021), could provide a quantitative basis for determining the subsets $A_K$ to consider.

## 5 CONCLUSION

This paper proposes a novel *Random-Set Neural Network (RS-NN)* for uncertainty estimation in classification predicting belief functions. Our random-set representation is a foundational approach that acts as a versatile wrapper, applicable to any model architecture and classification task (e.g, text). To our knowledge, this concept, along with budgeting for optimal selection of sets, is unprecedented. RS-NN outperforms state-of-the-art uncertainty estimation models and the standard CNN in performance and out-of-distribution OoD tests, and scales seamlessly to large-scale architectures and datasets. Given such results, our approach exhibits significant potential impact for safety-critical applications such as medical diagnostics and autonomous driving, where uncertainty estimation and OoD detection is crucial. Future work will explore alternative methods for ensuring the validity of belief functions without mass regularisation terms, such as integrating Semantic Probabilistic Layers (Ahmed et al., 2022) to enforce logical constraints in probabilistic predictions, or applying softmax over the computed masses to guarantee non-negativity (see §E.6). A parameter-level representation of RS-NN and the extension of this concept to regression will also be explored.

## REPRODUCIBILITY STATEMENT

To ensure reproducibility, we provide detailed information on the datasets, baseline models, and the training procedure for RS-NN and all baselines used in our experiments in Sec. 4.1, under the subheadings 'Datasets' and 'Baselines, backbones, and training details' respectively. Additional hyperparameters for RS-NN are provided under 'Training the RS-NN' (Sec. 4.1), and for baseline models in §D. For convenience, we summarize the training hyperparameters for all models in Tab. 5. Additionally, the training times for each model are included in Tab. 6 (§D). The code for training RS-NN (`train.ipynb`), including pre-trained models (for faster evaluation), evaluation script (`eval.ipynb`), and configuration files are provided as supplementary materials. We also provide a `.yml` file to set up the environment to run the experiments.

## ACKNOWLEDGEMENT

This work has received funding from the European Union's Horizon 2020 Research and Innovation program under Grant Agreement No. 964505 (E-pi).

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

APPENDIX: RANDOM-SET NEURAL NETWORKS

This appendix provides a comprehensive overview of Random-Set Neural Networks (RS-NN), including theoretical underpinnings (§A), related work (§B), algorithms (§C), implementation details (§D), additional experimental results (§E) which addresses critical questions to elucidate the functionality and robustness of RS-NN, and an application to text classification (§F).

# A    THEORETICAL EXPLANATION OF RS-NN

In this section, we provide an overview of belief functions (§A.1), describe the RS-NN learning mechanism (§A.2), and show the relationship between pignistic and credal predictions (§A.3), and provide statistical guarantees for RS-NN by applying conformal prediction to the predicted pignistic probabilities (§A.4).

## A.1    CLASSICAL PROBABILITIES VS BELIEF FUNCTIONS

In this section, we explain the difference between classical probability measures and non-additive measures, such as random sets. Random sets (Matheron, 1975; Kendall, 1974; Nguyen, 1978; Molchanov, 2005) are *non-additive* measures (i.e., the additivity property does not hold). Their additional degrees of freedom allow to express the 'epistemic' uncertainty about probability values themselves. In fact, an entire field on 'imprecise probabilities' (Augustin et al., 2014; Walley, 2000) exist which deals with these kinds of measures.

In a standard classification task, where each input is classified into one of several mutually exclusive classes, the classical probability framework does assume that each class is distinct and independent. The output of a softmax model provides the probabilities for each class, and these probabilities sum to 1. This approach works well when you are confident about the classification, for you have sufficient evidence to assign a clear probability to each class. However, it does not capture the uncertainty or ambiguity that exists when the evidence (as provided by the training set) is insufficient (e.g., because you did not sample similar points at training time, or because the test data is affected by distribution shift).

In Sec. 1 of the paper, we highlight that relying solely on confidence measures (like softmax probabilities) is insufficient for a comprehensive analysis of model predictions. Fig. 2 shows that standard neural networks are often overconfident (Nguyen et al., 2015) even for misclassifications. Our approach uses random sets (which, in the finite case, assume the name of belief functions), as detailed in Sec. 2, to offer a robust framework for assessing prediction reliability.

Random sets, as the name suggest, assign probability mass values to sets *independently*, and are therefore more general than classical probability measures (which are a special case of random set). As a result, belief functions offer a way to handle uncertainty and make decisions when information is incomplete or ambiguous. While in classical probability, the sum of probabilities for individual events and their unions adheres to strict additive rules, belief functions relax these constraints to better capture uncertainty. In particular, the belief in the union of two hypotheses, Bel({A,B}), can be greater than the sum of the beliefs in the individual hypotheses, Bel({A}) + Bel({B}), i.e, Bel({A}) + Bel({B}) $\leq$ Bel({A,B}). This inequality captures the idea that, in some situations, the available evidence may support the true class being in the set $\{A, B\}$, without being enough to distinguish between the two alternatives. If Bel({A,B}) > Bel({A}) + Bel({B}), it indicates that the model has significant uncertainty or confusion about distinguishing between these classes for a particular input.

Unlike classical probability theory, belief functions operate in a higher space than probability vectors, i.e, it operates in a framework that generalizes classical probability theory. Suppose we know with 100% certainty that an event is either A or B. In a classical probability setting, we might say P(A) = 0.5 and P(B) = 0.5, or we might assign different probabilities to A and B if we have some reason to favor one over the other. With belief functions, we can represent the same scenario differently. We could assign a belief value of 1 to the set {A, B} and 0 to both A and B individually (Bel({A}) = Bel({B}) = 0, Bel({A, B}) = 1). This means that while we know the outcome must be either A or B, we are entirely uncertain about which one it is. Alternatively, we could have Bel({A}) = 0.5, Bel({B}) = 0.5, Bel({A, B}) = 1, reflecting a different state of knowledge or evidence, where we're

somewhat more informed but still not fully certain. In the behavioural interpretation of probability (Walley, 2000), the belief value of an event A is the upper bound to the price one is willing to pay for betting on an outcome A.

Consider the following example (Cuzzolin, 2020): Suppose there is a murder, and three people—Peter, John, and Mary—are suspects. Our hypothesis space is therefore $\Theta = \{$Peter, John, Mary$\}$. A witness testifies that the person he saw was a man, supporting the proposition A = {Peter, John} $\subseteq \Theta$. However, the witness was tested, and the machine reported a 20% chance that he was drunk when he gave his testimony. As a result, we assign 80% belief to the proposition A, representing our uncertainty about whether any of the two could be the murderer.

In classical probability, Kolmogorov's additive probability theory (Kolmogorov & Bharucha-Reid, 2018) forces us to specify support for *individual outcomes*, i.e. to distribute this 80% probability between Peter and John individually, even when the evidence (our data) supports *set propositions*. This is unreasonable – an artificial constraint due to a mathematical model that is not general enough. In the example, we have do not have enough evidence to assign this 80% probability to either Peter or John, nor information on how to distribute it amongst them. The cause is the additivity constraint that probability measures are subject to. This constraint reflects a broader limitation of measure-theoretical probability, which, while effective in many scenarios, struggles with second-order uncertainties and incomplete information. Therefore, we have several other methods to estimate uncertainty, such as the Bayesian method, or using degrees of belief, such as Evidential and belief function methods (see Sensoy et al. (2018), Sec. 3).

This means that *belief functions allow us to model situations where we have uncertainty not just about which class an input belongs to, but also about whether we have enough evidence to distinguish between certain classes*. This added flexibility is what makes belief functions suitable for single-label classification tasks.

## A.2 RS-NN LEARNING MECHANISM

Here we wish to expand on the rationale behind our choices for ground-truth representation **bel**, loss function and training, and the way in which RS-NN learns sets of outcomes.

Recall from Sec. 3.1 that the ground-truth for a RS-NN is the belief-encoded vector $\mathbf{bel} = \{Bel(A), A \in \mathbb{P}(\mathcal{C})\}$, where $Bel(A)$ is the belief function of a focal set $A$ in the power set $\mathbb{P}$ of classes $\mathcal{C}$. In our method, $Bel(A)$ is 1 if the true class is in subset $A$ and 0 otherwise. Consequently, the belief-encoded ground truth **bel** will include multiple occurrences of 1. For instance, in a digit-classification task such as MNIST, if $\{3\}$ is the true class, the belief-encoded ground truth would contain 1s in all sets where $\{3\}$ is present, such as $\{1, 3\}, \{0, 3\}, \{1, 2, 3\}$, and so forth, 0 for sets not containing $\{3\}$ at all, such as $\{0, 1\}, \{1, 2\}, \{0, 1, 2\}$, etc.

Since we assign precise labels for belief containing different focal sets, our model begins by penalizing predictions that differ from the observed label in the same manner, regardless of the set's composition or relationship to the true label. Using MNIST as an example, this means that the loss incurred for predicting label $\{3\}$ is equivalent to predicting label $\{3, 7\}$, as both sets contain the true label. This equivalence in losses might seem counterintuitive at first glance. However, despite the identical loss values, the probabilities output by the sigmoid activation function will vary due to differences in the input logit values for each label. For instance, if the correct label is $\{3\}$, the loss for predicting $\{3\}$ or $\{3, 7\}$ would be the same. Still, the loss for predicting $\{7\}$ would differ, allowing the model to discern the set structure during training.

During the training process, the model learns to capture and understand these relationships by observing patterns and dependencies in the training data. As the model optimizes its parameters based on the training objective (e.g., minimizing the loss function), it gradually adjusts its internal representations to better reflect these relationships. We use as the basis for our loss function Binary cross-entropy with sigmoid activation. It allows the model to predict the presence or absence of each label separately as a binary classification problem, producing probabilities between 0 and 1 for each class. While the model is unaware that it is learning for sets of outcomes, we leverage this technique to extract mass functions and pignistic predictions from the learnt belief functions. By re-distributing masses to original classes, we obtain the best pignistic predictions that are often more accurate than standard CNN predictions, attributing to the higher accuracy in Tab. 1.

## A.3 Pignistic probability and credal prediction

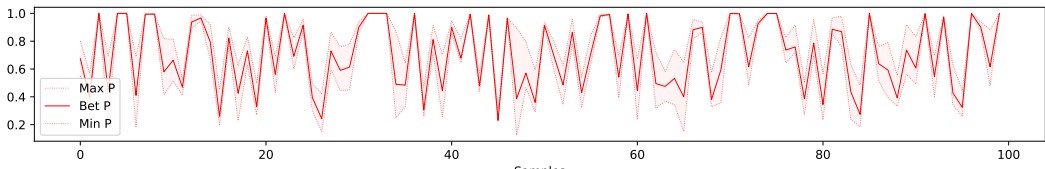

Figure 7: Upper and lower bounds (Eq. 9), in dotted red, to the probability of the predicted most likely class according to the pignistic prediction (in solid red) for 100 samples of CIFAR-10.

As they have belief values as lower bounds, credal sets induced by belief functions have a peculiar shape (see Fig. 4, right, for a case with 3 classes). Their vertices are induced by the permutations of the elements of the sample space (for us, the set of classes) (Chateauneuf & Jaffray, 1989; Cuzzolin, 2008). Given one such permutation, e.g., $(c_2, c_4, c_1, c_3)$ for a set of 4 classes, the corresponding probability vector (vertex of the credal prediction) assigns to each class the mass of all the focal sets containing it, *but* not containing any class preceding it in the permutation order (Wallner, 2005).

For additional illustration, the lower and upper bounds (Eq. 9) to the probability of the top predicted class are plotted in Fig. 7, together with the pignistic probability, for 100 samples of CIFAR-10. The bounds Eq. 9 can be efficiently computed from the finite number of vertices of the credal prediction ((Cuzzolin, 2008), Fig. 4).

## A.4 Statistical guarantees

To complement the extensive discussion in the main paper, we provide here an in-depth discussion on how statistical guarantees can be provided in an RS-NN framework.

**Plugging RS-NN into conformal learning.** Indeed, RS-NN can be employed as the 'underlying model' in an inductive conformal learning framework[3], which builds an empirical cumulative distribution of the 'non-conformity' scores of a set of calibration samples, and at test time outputs the set of labels whose empirical CDF is above a desired significance level $\epsilon$ (e.g., 95%).

Given a test input $x$ and the associated predictive belief function $\hat{Bel}(c|x)$ (the output of RS-NN), we could, for instance, set as non-conformity score

$$s(x, c) \doteq 1 - \hat{Pl}(c|x) = \hat{Bel}(\mathcal{C} \setminus \{c\}|x) \tag{10}$$

(i.e., a label $c$ is 'non-conformal' if its predicted *plausibility*, which is defined as $Pl(A) = 1 - Bel(\Theta \setminus A)$ and has the semantic of an upper probability bound (Shafer, 1976), is low), and compute predictive regions in the usual way:

$$\Gamma(x) = \{c \in \mathcal{C} : p^c > \epsilon\},$$

where

$$p^c = \frac{|(x_j, c_j) : s(x_j, c_j) > s(x, c)|}{q + 1} + u \cdot \frac{|(x_j, c_j) : s(x_j, c_j) = s(x, c)|}{q + 1},$$

$(x_j, c_j)$ is the $j$-th calibration point, $q$ is the number of calibration points, and $u \sim \mathcal{U}(0, 1)$ (the uniform distribution on the interval $(0, 1)$).

**Experiments on conformal guarantees.** Thus, we conducted experiments on CIFAR-10 dataset to provide conformal prediction guarantees with non-conformity scores as given in Eq. 10. The goal is to assess the reliability of classification predictions by determining a threshold of non-conformity scores that covers 95% of the data.

To achieve this, non-conformity scores are obtained for all classes, and a threshold is calculated such that it contains 95% of the data. This threshold determines the coverage of the classification, which refers to the proportion of predictions that actually contain the true outcome.

---

[3]https://cml.rhul.ac.uk/copa2017/presentations/CP_Tutorial_2017.pdf

This prediction threshold is determined by finding the percentile corresponding to $1 - \alpha$, where $\alpha$ is set to $0.05$ to achieve 95% coverage. Fig. 9 shows the non-conformity scores for all the calibration data, $j = 1000$ (Angelopoulos & Bates, 2021) (using the following split: 40000:10000:9000:1000 training, testing, validation and calibration data points respectively for CIFAR-10), with a cut-off threshold that ensures 95% coverage.

The conformal prediction sets for two sample inputs are shown in Fig. 8, along with their predicted probabilities.

```
Prediction sets: {'cat', 'horse', 'truck'}      Prediction sets: {'cat', 'dog'}
[0.20296144] [0.54583361] [0.24168862]          [0.49130123] [0.41791199]
True label: horse                               True label: cat
```

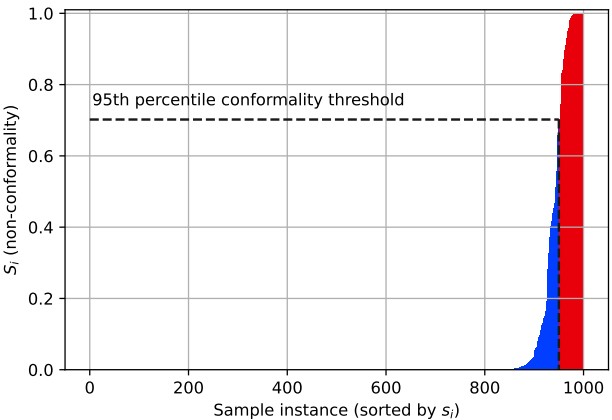

Figure 8: Conformal prediction sets by RS-NN on the CIFAR-10 dataset. The predicted probabilities for each class is shown.

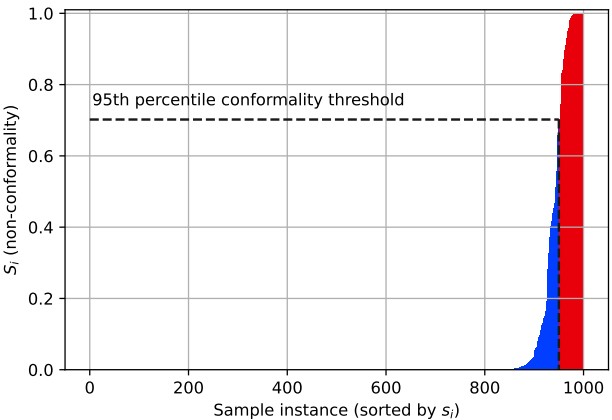

Figure 9: Conformal prediction threshold for CIFAR-10 indicating the boundary beyond which predictions are considered non-conformal.

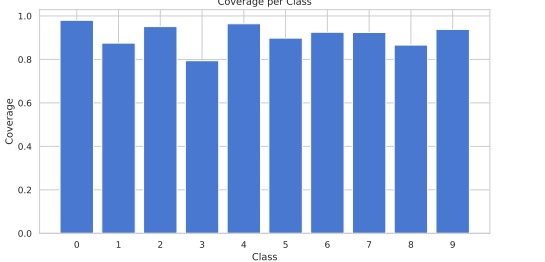

Figure 10: Coverage per class of RS-NN for CIFAR-10 dataset.

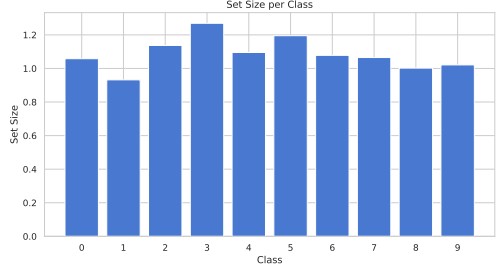

Figure 11: Average size of the predictive sets generated for each class of CIFAR-10.

Figs. 10 and 11 show the coverage and average set size for each class on the test dataset. Coverage is the proportion of prediction sets that actually contain the true outcome, and average set size is the

average number of predicted classes per instance. Higher numbers represent more overlap between the classification regions of different classes.

Summarizing the results, (1) the coverage averages across classes over the desired threshold. This indicates that the conformal prediction method is providing reliable prediction intervals that contain the true class label with the specified confidence level; (2) The spread or variability in set sizes across different samples provides insights into how well the prediction sets adapt to the difficulty of the samples. Ideally, the prediction sets should dynamically adjust based on the difficulty of each example, with larger sets for more challenging inputs and smaller sets for easier ones. Fig. 11 shows that the average set size for classes 1, 2, 3, 4, and 6 is considerably larger than for the rest, indicating that the model found samples from these as challenging inputs.

**Alternative approaches to statistical guarantees.** In the future we plan to explore the possibility of generalising the empirical CDF at the basis of conformal learning in a belief function / random set representation, aiming to retain statistical guarantees. Given the complexity of this enterprise, we are working towards this in a separate paper.

Finally, an intriguing alternative approach (and one that is more achievable in the short term) is the study of confidence intervals in a belief functions representation, such as the one we employ in RS-NNs. In fact, recent studies have been looking at extending the notion of confidence interval (`https://en.wikipedia.org/wiki/Confidence_interval`) to belief functions, under the name of *confidence structures* (`https://hal.science/hal-01576356v3`), which generalise standard confidence distributions and generate "frequency-calibrated" belief functions.

Liu and Martin, in particular, have developed an Inferential Model (IM) approach which produces belief functions with well-defined frequentist properties (Martin & Liu, 2013; 2015). An alternative approach relies on the notion of "predictive" belief function (Denœux, 2006), which, under repeated sampling, is less committed than the true probability distribution of interest with some prescribed probability.

## B    RELATED WORK

Researchers have recently taken some tentative steps to model uncertainty in machine learning and artificial intelligence. Scientists have adopted measures of epistemic uncertainty (Kendall & Gal, 2017) to refuse or delay making a decision (Geifman & El-Yaniv, 2017), take actions specifically directed at reducing uncertainty (as in active learning (Aggarwal et al., 2014)), or exclude highly-uncertain data at decision making time (Kendall & Gal, 2017; Manchingal et al., 2025).

**Imprecise probability.** Significant work has been done in a credal set setting. Notably, (Zaffalon, 2002; Corani & Zaffalon, 2008) have proposed the Naive Credal Classifier (NCC) as an extension of the naive Bayes classifier to credal sets, where imprecise probabilities are included in models in the form of sets of classes. (Antonucci & Corani, 2017) have presented graphical models which generalise NCCs to multilabel data. Expected utility maximisation algorithms, such as Bayes-optimal prediction (Mortier et al., 2021), and classification with reject option for risk aversion (Nguyen et al., 2018) are also based on set-valued predictions.

Classification with partial data has been studied by various authors (Vannoorenberghe & Smets, 2005), while a large number of papers have been published on decision trees in the belief function framework (Elouedi1 et al., 2000). Significant work in the neural network area was conducted in (Denoeux, 2000) and (Fay et al., 2006). Classification approaches based on rough sets were proposed in (Trabelsi et al., 2011). Ensemble classification (Burger et al., 2006) is another area in which uncertainty theory has been quite impactful, as the problem is one of combining the results of different classifiers. Important contributions have been made by (Xu et al., 1992) and (Rogova, 2008). Regression, on the other hand, has only recently been considered by belief theorists (Laanaya et al., 2010; Gong & Cuzzolin, 2018; Denœux, 2022). For tasks like image classification in large datasets, most of these approaches have low performance metrics including training time and test accuracy.

**Evidential approaches**. Within a proper epistemic setting, a significant amount of work has been done by Denoeux and co-authors, and Liu et al. (2012), on unsupervised learning and clustering in particular in the belief function framework. Quite a lot of work has been done on ensemble classifica-

tion in the evidential framework (Xu et al., 1992) (in particular for neural networks (Rogova, 2008)), decision trees (Elouedi et al., Madrid, 2000), K-nearest neighbour classifiers (Denœux, 2008), and more recently on evidential deep learning classifiers able to quantify uncertainty (Tong et al., 2021). Tong et al.(Tong et al., 2021) proposes a convolutional neural network based on Dempster-Shafer theory called the evidential deep-classifier employs utility functions from decision theory to assign utilities on mass functions derived from input features to produce set-valued observations. Another recent approach by Sensoy et al.(Sensoy et al., 2018) proposes an evidential deep learning classifier to estimate second-order uncertainty in the Dirichlet representation. This work is based on subjective logic and learning to form subjective opinions by minimizing the Kullback-Leibler divergence over a uniform Dirichlet distribution.

These methods represent predictions as Dirichlet distributions, as explored in various works over the last few years (Sensoy et al., 2018; Malinin & Gales, 2018; 2019; Malinin et al., 2019; Charpentier et al., 2020). However, many commonly used loss functions for these networks are flawed, as they fail to ensure that epistemic uncertainty diminishes with more data, violating basic asymptotic assumptions (Bengs et al., 2022). Additionally, some approaches require out-of-distribution (OoD) data for training, which may not always be available and doesn't guarantee robustness against all types of OoD data. Studies (Ulmer et al., 2023) have shown that OoD detection degrades in certain models under adversarial conditions, and even techniques like normalizing flows (NFs) in posterior networks (Charpentier et al., 2020), while effective, can struggle with OoD data when relying on learned features (Kopetzki et al., 2021; Stadler et al., 2021).

**Conformal prediction**. Conformal prediction (Vovk et al., 2005) provides a framework for estimating uncertainty (Shafer & Vovk, 2008) by applying a threshold on the error the model can make to produce prediction sets, irrespective of the underlying prediction problem. Different variants of conformal predictors are described in papers by Saunders et al. (Saunders et al., 1999), Nouretdinov et al. (Nouretdinov et al., 2001), Proedrou et al. (Proedrou et al., 2002) and Papadopoulos et al. (Papadopoulos et al., 2008).

Since the computational inefficiency of conformal predictors posed a problem for their use in neural networks, Inductive Conformal Predictors (ICPs) were proposed by Papadopoulos et al. (Papadopoulos et al., 2002a) (Papadopoulos et al., 2002b). Venn Predictors (Vovk et al., 2003), cross-conformal predictors (Vovk, 2012) and Venn-Abers predictors (Vovk & Petej, 2012) were introduced in distribution-free uncertainty quantification using conformal learning. As we show above, RS-NN is compatible with conformal prediction, as a possible underlying model. In the future we plan to extend our random-set representation to conformal learning, by replacing cumulative distribution functions with random sets.

**Deterministic uncertainty quantification.** Deterministic Uncertainty Quantification (DUQ) methods estimate epistemic uncertainty by analyzing latent representations or using distance-sensitive functions instead of softmax (Alemi et al., 2018; Wu & Goodman, 2020; Liu et al., 2020; Mukhoti et al., 2021; Van Amersfoort et al., 2020). These methods have been applied to areas like object detection (Gasperini et al., 2021) but focus mainly on OoD detection, overlooking calibration — how well uncertainty reflects model performance under shifting distributions. Calibration is crucial for safe deployment but remains underexplored, as DUQs are typically tested only on simple tasks and datasets, leaving their effectiveness in more complex scenarios unproven (Postels et al., 2021).

Uncertainty estimates are poorly calibrated under distributional shifts, especially when compared to scalable Bayesian methods. Regularization techniques based on feature space distances, such as bi-Lipschitz regularization (used by most of these models), do not effectively improve OoD detection or calibration (Postels et al., 2021). This is due to the limitations of distance metrics in high-dimensional data, such as images. Additionally, DUQ methods differ from RS-NN by modeling epistemic uncertainty in the input space rather than the prediction space. They assess uncertainty based on whether the model's input is an iD or OoD sample. Because of this difference in how epistemic uncertainty is modeled, we cannot directly compare RS-NN to DUQ methods.

## C   ALGORITHMS

This section outlines the algorithms integral to the implementation and evaluation of budgeting (§C.1), expected calibration error (ECE) (§C.2), area under the receiver operating characteristic curve (AUROC) and area under the precision-recall curve (AUPRC) (§C.3).

## C.1 ALGORITHM FOR BUDGETING

For RS-NN with $N$ classes, generating $2^N$ outputs is computationally infeasible due to exponential complexity. Instead, we choose $K$ relevant subsets ($A_K$ focal sets) from the $2^N$ possibilities.

To obtain these $K$ focal subsets, we extract feature vectors from the penultimate layer of a trained standard CNN with $N$ outputs. We then apply t-SNE for dimensionality reduction to 3 dimensions. Note that our approach is agnostic, as t-SNE could be replaced with any other dimensionality reduction technique, including autoencoders.

Next, we fit a Gaussian Mixture Model (GMM) to the reduced feature vectors of each class. Using the eigenvectors and eigenvalues of the covariance matrix $\Sigma_c$ and the mean vector $\mu_c$ for each class $c$, we define an ellipsoid covering 95% of the data. The lengths of the ellipsoid's principal axes are computed as $length_{c,i} = 2\sqrt{7.815\lambda_i}$, where $\lambda_i$ is the $i^{th}$ eigenvalue. The scalar $7.815$ corresponds to a 95% confidence interval for a chi-square distribution with 3 degrees of freedom. The class ellipsoids are plotted in a 3D space and the overlap of each subset in the power set of $N$ is computed. As it is computationally infeasible to compute overlap for all $2^N$ subsets, we start doing so from cardinality 2 and use early stopping when increasing the cardinality further does not alter the list of most-overlapping sets of classes. We choose the top-$K$ subsets ($A_K$) with the highest overlapping ratio, computed as the intersection over union for each subset.

---

**Algorithm 1** Budgeting Algorithm

---

1: **Input:** $\mathcal{D}$ – Training data with $N$ classes, $\mathcal{C}$ – The set of classes, $K$ – Number of non-singleton focal sets
2: **Output:** $\mathcal{O}$ – Set containing $N + K$ focal sets
3: *Initialization*
4: Extract feature vectors using a trained CNN
5: Apply t-SNE for dimensionality reduction to 3 dimensions
6: **for** each class $c$ **do**
7:     Fit GMM to the reduced feature vectors for that class to obtain $\mu_c$ and $\Sigma_c$
8:     Define an ellipsoid covering 95% data using $\mu_c$ and $\Sigma_c$
9: **end for**
10: most_overlapping_sets $\leftarrow$ Initialize an empty list for non-singleton focal sets $A_1, \ldots, A_K$
11: Set $current\_cardinality \leftarrow 2$
12: **while** $current\_cardinality \leq N$ **do**
13:     *Compute overlaps for subsets of cardinality current_cardinality*
14:     $\text{overlap}(A) = \frac{\cap_{c \in A} A^c}{\cup_{c \in A} A^c}$
15:     *Select top-K subsets with highest overlap*
16:     Update most_overlapping_sets
17:     **if** no change in most_overlapping_sets **then**
18:         **break**
19:     **end if**
20:     $current\_cardinality \leftarrow current\_cardinality + 1$
21: **end while**
22: *Combine selected non-singleton focal sets with $N$ singleton sets*
23: $\mathcal{O} \leftarrow \mathcal{C} \cup \{A_1, \ldots, A_K\}$
24: **return** $\mathcal{O}$

---

## C.2 ALGORITHM FOR ECE

Expected Calibration Error (ECE) is computed by comparing the average confidence and accuracy within each bin. The steps involved are as follows:

1. For each bin, calculate the absolute difference between the average confidence and the accuracy.

2. Weight each difference by the proportion of instances in that bin compared to the total number of instances.

3. Sum up these weighted differences across all bins to get the final ECE value.

---

**Algorithm 2** Expected Calibration Error (ECE)

---

**Require:** *confidences*: List or array of confidence scores predicted by the model *predictions*: List or array of predicted class labels *true_labels*: List or array of true class labels $B$: Number of bins for binning confidence scores

**Ensure:** *ECE*: Expected Calibration Error

1: **Step 1: Normalize Confidences** Normalize the confidence scores to ensure they are in the range [0, 1].
2: **Step 2: Binning** Divide the confidence scores into *num_bins* bins.
3: **Step 3: Initialize Arrays** Initialize arrays *bin_accuracy*, *bin_confidence*, and *weights* with zeros.
4: **Step 4: Populate Arrays**
5: **for** each bin **do**
6:     Identify instances falling into the bin
7:     Calculate mean accuracy and mean confidence within the bin
8:     Update *bin_accuracy*, *bin_confidence*, and *weights*
9: **end for**
10: **Step 5: Calculate ECE** Calculate the Expected Calibration Error using the populated arrays.

$$ECE = \sum_{i=1}^{B} |(bin\_accuracy_i - bin\_confidence_i)| \times weights_i$$

---

The formula for ECE is given by:

$$\text{ECE} = \sum_{i=1}^{B} |(\text{Accuracy}_i - \text{Confidence}_i)| \times \text{Weight}_i$$

where:

- $\text{Accuracy}_i$ is the accuracy within the $i$-th bin.
- $\text{Confidence}_i$ is the average confidence within the $i$-th bin.
- $\text{Weight}_i$ is the proportion of instances in the $i$-th bin compared to the total number of instances.
- $B$ is the total number of bins.

The final ECE value represents how much the model's estimated probabilities differ from the true (observed) probabilities. A low ECE indicates well-calibrated probabilistic predictions, demonstrating that the model's confidence scores align closely with the actual likelihood of events. RS-NN exhibits the lowest ECE as shown in Tab 2.

### C.3 ALGORITHM FOR AUROC, AUPRC

Out-of-distribution (OoD) detection involves the identification of data points that deviate from the in-distribution (iD) data on which a model was trained. This process relies on assessing the model's uncertainty, particularly its epistemic uncertainty, which reflects the model's lack of knowledge or confidence in making predictions. When exposed to OoD data, which differs significantly from the training data, the model tends to exhibit higher epistemic uncertainty.

AUROC (Area Under the Receiver Operating Characteristic Curve) and AUPRC (Area Under the Precision-Recall Curve) are evaluation metrics commonly used to assess the performance of binary classification models, providing insights into their ability to distinguish between positive and negative instances. In the context of evaluating uncertainty estimation in machine learning models, these metrics quantify how well the model separates iD samples from OoD samples. AUROC is derived from the Receiver Operating Characteristic (ROC) curve, which illustrates the trade-off between True Positive Rate (TPR) and False Positive Rate (FPR) across various classification thresholds.

The AUROC value represents the area under this curve and ranges from 0 to 1, with higher values indicating better discriminative performance. AUPRC is based on the Precision-Recall curve, which

---

**Algorithm 3** Algorithm for AUROC, AUPRC

---

**Require:** *uncertainty_iid*: Uncertainty scores for in-distribution samples. *uncertainty_ood*: Uncertainty scores for out-of-distribution samples.
**Ensure:** (fpr, tpr, thresholds): ROC curve metrics.
 1: (precision, recall, prc_thresholds): Precision-Recall curve metrics.
 2: auroc: Area Under the Receiver Operating Characteristic curve.
 3: auprc: Area Under the Precision-Recall curve.
 4: **Step 1: Concatenate uncertainties;**
 5: uncertainties ← concatenate(uncertainty_iid, uncertainty_ood)
 6: **Step 2: Create and combine labels;**
 7: in_labels ← zeros(uncertainty_iid.shape[0]) ood_labels ← ones(uncertainty_ood.shape[0]) labels ← concatenate(in_labels, ood_labels)
 8: **Step 4: Calculate ROC curve;**
 9: (fpr, tpr, thresholds) ← roc_curve(labels, uncertainties)
10: **Step 5: Calculate AUROC;**
11: auroc ← roc_auc_score(labels, uncertainties)
12: **Step 6: Calculate Precision-Recall curve;**
13: (precision, recall, prc_thresholds) ← precision_recall_curve(labels, uncertainties)
14: **Step 7: Calculate AUPRC;**
15: auprc ← average_precision_score(labels, uncertainties)

---

plots Precision against Recall at different classification thresholds. Precision measures the accuracy of positive predictions, while Recall quantifies the ability to capture all positive instances. AUPRC calculates the area under this curve and provides a complementary perspective, particularly valuable when dealing with imbalanced datasets.

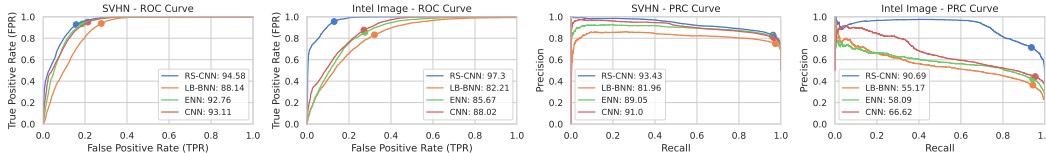

Figure 12: Receiver Operating Characteristic (ROC) and Precision-Recall Characteristic (PRC) curves for RS-NN, LB-BNN, ENN, and CNN evaluated on the SVHN and Intel Image OoD datasets for CIFAR-10.

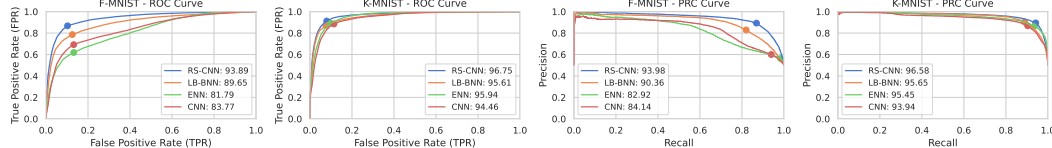

Figure 13: Receiver Operating Characteristic (ROC) and Precision-Recall Characteristic (PRC) curves for RS-NN, LB-BNN, ENN, and CNN evaluated on the SVHN and Intel Image OoD datasets for MNIST.

Figs. 12 and 13 plot each model's performance, illustrating the trade-offs between True Positive Rate and False Positive Rate (ROC curve), Precision and Recall (PRC curve) for CIFAR-10 (Fig. 12) and MNIST (Fig. 13). The left two plots depict the AUROC curves, where the blue curve representing RS-NN outperforms others, indicating superior discrimination between true positive and false positive rates. Similarly, on the two right plots displaying Precision-Recall Curve (PRC), RS-NN exhibits the highest curve, emphasizing its precision and recall performance. These results showcase RS-NN's effectiveness in distinguishing in-distribution and out-of-distribution samples.

In real-world scenarios, encountering OoD instances is inevitable, making reliable OoD detection essential for safety-critical applications to avoid erroneous decisions on unfamiliar data. Recognising unfamiliar data signals the model about situations beyond its training, allowing it to acknowledge its own limitations and ignorance, which in turn enhances its uncertainty estimation.

## D    IMPLEMENTATION DETAILS

Laplace Bridge Bayesian approximation (LB-BNN) (Hobbhahn et al., 2022) uses the Laplace Bridge to map efficiently between Gaussian and Dirichlet distributions and enhance computational efficiency. In our experiments, we obtain multiple samples from the LB-BNN posterior and average these samples using Bayesian Model Averaging (BMA). Function-Space Variational Inference (FSVI) (Rudner et al., 2022) utilizes variational inference to derive an approximation of the posterior distribution over the function space. FSVI proposes approximating distributions over functions as Gaussian by linearizing their mean parameters and derived a tractable and well-defined variational posterior. Deep Ensembles (DEs) (Lakshminarayanan et al., 2017) and Epistemic Neural Networks (ENNs) (Osband et al., 2024) are ensemble methods that train ensembles of models to efficiently estimate uncertainty. The mean and variance of ensemble predictions are calculated and the mean is softmaxed to obtain predictions from DE and ENN. The training hyperparameters for all models are outlined in Tab. 5.

Table 5: Training Hyperparameters for All Models

| Parameter | Value |
|---|---|
| **Architecture** | ResNet50 (excluding top classification layer) |
| **Additional Dense Layers** | 1024 and 512 neurons (ReLU activation) |
| **Output Layer Activation (RS-NN)** | Sigmoid (multi-label classification) |
| **Output Layer Activation (Other Models)** | Softmax (multi-class classification) |
| **Learning Rate** | Initial: 1e-3 |
| **Learning Rate Scheduler** | Decrease by 0.1 at epochs 80, 120, 160, 180 |
| **Training Epochs** | 200 |
| **Batch Size** | 128 |
| **Optimizer** | RS-NN: Adam, LB-BNN: Adam, ENN: Adam, DE: Adam, FSVI: SGD |
| **Training Dataset Sizes** | CIFAR-10: 40,000
CIFAR-100: 40,000
MNIST: 50,000
Intel Image: 13,934
ImageNet: 1,172,498 |
| **Testig Dataset Sizes** | CIFAR-10: 10,000
CIFAR-100: 10,000
MNIST: 10,000
Intel Image: 3,000
ImageNet: 2,000 |
| **Testing Samples for OoD Datasets** | 10,000 (except Intel Image: 3,000) |
| **Input Image Size** | $224 \times 224$ |
| **Data Augmentation** | Random horizontal/vertical shifts (magnitude 0.1), horizontal flips |
| **GPU** | NVIDIA A100 80GB |

We report the training times (in minutes) for all models on the CIFAR-10 dataset in Tab. 6. Tab. 6 shows that FSVI has the longest training time, followed by RNN and DE. Compared to LB-BNN, RS-NN has an additional 5 minutes to the training duration, while CNN has the shortest training time. However, CNNs do not provide uncertainty estimation, and RS-NN has more reliable uncertainty and OoD metrics in comparison to LB-BNN.

The significant training time required for Deep Ensembles (DE) often does not justify their uncertainty estimates (Abe et al., 2022; Rahaman et al., 2021). Despite the computational cost, DE is sometimes only as good as other uncertainty estimation models, which can be trained much faster. Other approaches, like methods with calibration techniques (e.g., temperature scaling or Bayesian approaches), can provide similar performance without the substantial overhead that DE entails.

Table 6: Training time (min) for all models on the CIFAR-10 dataset.

| | RS-NN | LB-BNN | FSVI | DE | ENN | CNN |
|---|---|---|---|---|---|---|
| Training time (min) | 113.23 | 107.900 | 1518.35 | 426.66 | 712.302 | **85.333** |

As mentioned in Sec. 4.2, the results for FSVI are different than what was reported in (Rudner et al., 2022). The lower performance with ResNet-50 in our experiments may be attributed to the model being optimized specifically for ResNet-18. We noticed the unexpectedly low results and adhered to their suggested optimizer, SGD, instead of Adam (used in all other models), to achieve optimal performance, as mentioned in Tab. 5. Additionally, to enhance performance, we also incorporated the context points from ImageNet and CIFAR-100 as recommended in Rudner et al. (2022). Furthermore, we observed that FSVI uses more extensive data augmentation than our standardized setup. For instance, their data augmentation includes random crops and higher magnitude random flips (0.5), while we only used random horizontal/vertical flips of magnitude 0.1, potentially contributing to the performance gap. While increasing model capacity by using ResNet-50 (25.6M parameters) over ResNet-18 (11.7M parameters) should theoretically improve performance, this is not always the case in practice. ResNet-50 requires more careful optimization and is more prone to overfitting, particularly when used with smaller datasets or suboptimal augmentation. This explains why ResNet-50 did not consistently outperform ResNet-18 in our implementation of the compared methods.

## E  ADDITIONAL EXPERIMENTAL RESULTS

In this section, we will address the following questions based on our experimental findings:

- **Q1:** Why are most models not able to differentiate between ImageNet and ImageNet-O in terms of entropy?

- **Q2:** Why does RS-NN achieve better accuracy than CNN? What is the overconfidence problem in CNN?

- **Q3:** How does RS-NN handle perturbed data (noisy, rotated)?

- **Q4:** What empirical evidence supports the claim that RS-NN is more robust against adversarial attacks than traditional CNNs?

- **Q5:** Why is the credal set considered a good measure of epistemic uncertainty?

- **Q6:** What is the importance of the mass regularizers in the loss function $\mathcal{L}_{RS}$ (Sec. 3.2)? Why is it important to properly tune the regularization parameters $\alpha$ and $\beta$, and what impact does this have on the model's performance?

- **Q7:** What are the effects of budgeting on RS-NN? How was the number of budgeted focal sets $K$ chosen? How does budgeted RS-NN differ from standard RS-NN with all the subsets in the power set?

- **Q8:** How does UMAP measure as a faster alternative to t-SNE for budgeting in RS-NN? How can the budgeting procedure be adapted for continuous data streams?

### E.1  IMAGENET VS IMAGENET-O

In Tab 2, the clear separation between iD and OoD entropy is evident for RS-NN, yet for ImageNet vs ImageNet-O, all other models struggle to distinguish between the iD vs OoD entropy and the OoD dataset almost falls within the standard deviation of the iD samples. Also, in Tab. 3, the credal set widths for CIFAR-10 vs SVHN/Intel Image and MNIST vs F-MNIST/K-MNIST are distinct, but ImageNet vs ImageNet-O is not too different.

To better understand this, it is important to consider how ImageNet-O is generated. ImageNet-O is a dataset of adversarially filtered examples for ImageNet out-of-distribution detectors. It is created by

removing all the ImageNet-1K samples from the full ImageNet-22K dataset. The remaining samples, which do not belong to ImageNet-1K classes, are classified by a trained model. The samples classified by the model as ImageNet-1K classes with high confidence becomes ImageNet-O. Hence, in general, it is difficult for models to make the iD vs OoD distinction as evident by uncertainty measures in Tab. 2.

We conducted experiments on Credal Set Width and Pignistic Entropy measures of RS-NN using a ViT-B-16 (Vision Transformer) backbone, as shown in Tab. 7. The pignistic entropy and credal set width show good distinction between iD and OoD measures, unlike the uncertainty measures from other baseline models in Tab. 2.

Table 7: Credal set width and entropy for RS-NN using ViT-B-16 model on ImageNet vs ImageNet-O datasets.

|  | Datasets | Credal Set Width | Pignistic Entropy |  | Softmax Entropy |
|---|---|---|---|---|---|
| RS-NN | ImageNet (iD) ($\downarrow$) | $0.1082 \pm 0.1926$ | $2.7401 \pm 2.3678$ | CNN | $2.3442 \pm 1.2243$ |
|  | ImageNet-O (OoD) ($\uparrow$) | $0.1948 \pm 0.2455$ | $4.4238 \pm 2.5744$ |  | $2.8619 \pm 1.5884$ |

Table 8: OoD detection and uncertainty estimation performance for iD vs OoD dataset: CIFAR-10 vs CIFAR-100.

| Dataset | Model | In-distribution Entropy ($\downarrow$) | Credal Set Width ($\downarrow$) | CIFAR-100 | | | |
|---|---|---|---|---|---|---|---|
|  |  |  |  | AUROC ($\uparrow$) | AUPRC ($\uparrow$) | Entropy ($\uparrow$) | Credal Set Width ($\uparrow$) |
| CIFAR-10 | RS-NN | $0.0802 \pm 0.297$ | $0.0061 \pm 0.039$ | 88.01 | 84.93 | $0.5778 \pm 0.729$ | $0.0721 \pm 0.165$ |

This is specifically due to the peculiar relation between ImageNet and ImageNet-O, and not a general problem with the models not being able to distinguish samples when they belong to the same image distribution.

To support our claim, we conducted experiments using CIFAR-10 (iD) vs CIFAR-100 (OoD) as shown in Tab. 8 below, and the results indicate that our model distinguishes between these two datasets well, as evident through the AUROC, AUPRC, entropy and credal set width. Therefore, the issue seems to be specific to ImageNet and ImageNet-O, likely due to the class imbalance in ImageNet. As per the general trend for ImageNet in Tab. 2, all models behave uncertainly for higher number of classes which makes the difference between iD and OoD less extreme. Although, RS-NN still performs better than other models on ImageNet.

### E.2 RS-NN VS CNNS: ACCURACY AND CONFIDENCE

Table 9: Standard CNN fails to perform well when tested on an noisy (rotated) sample of MNIST test data. The predicted results show how a standard CNN predicts the wrong class with a high confidence score (99.95%), while the RS-NN model predicts the correct class with 49.7% confidence and a high entropy of 2.1626.

| True Label = "8" | CNN Predictions | | RS-NN Predictions | | | | | |
|---|---|---|---|---|---|---|---|---|
|  |  |  | Belief values | | Mass values | | Pignistic Probability | |
|  | Class 6 | **0.9995** | {'6', '4', '8'} | 0.7853 | {'6', '8'} | 0.2079 | **8** | **0.4978** |
|  | Class 5 | 0.0002 | {'6', '8', '1'} | 0.7150 | {'9', '8', '1'} | 0.1999 | 6 | 0.2294 |
|  | Class 8 | 0.0001 | {'6', '8'} | 0.6357 | {'8'} | 0.1959 | 9 | 0.1002 |
|  | Class 0 | 1.4e-05 | {'9', '8'} | 0.5529 | {'8', '3'} | 0.0961 | 3 | 0.0502 |
|  | Class 4 | 1.1e-06 | {'8', '3'} | 0.4092 | {'6'} | 0.05147 | 4 | 0.0459 |
|  | **Entropy** 0.0061 | | **Entropy** 2.1626 | **Credal set width** 0.582 | | | | |

**Accuracy.** RS-NN consistently achieves higher accuracy than standard CNN (ResNet50) on all datasets as shown in Tab. 1. This appears to attest that, as RS-NN uses a random-set framework which does not require prior assumptions and is more data driven, it can better adapt and capture the inherent complexity of the data (as sets), contributing to its superior performance in test accuracy

across various datasets (Tab. 1). The training hyperparameters for RS-NN and CNN are the same, since the training procedure is quite similar.

**Confidence.** We further analyse how RS-NN overcomes the overconfidence problem in CNNs. Tab 9 shows an noisy example of a rotated MNIST image with a true label '8'. The standard CNN makes an incorrect prediction ('6') with $99.95\%$ confidence and a low entropy of $0.0061$, showcasing a noTab limitation in relying solely on confidence scores and entropy of predicted probabilities. RS-NN provides a correct prediction (class '8') with a confidence of $49.7\%$ and pignistic entropy of $2.1626$ and a credal set width of $0.482$. Crucially, the RS-NN's prediction process considers more than just the predicted class, taking into account both the confidence of the pignistic probability and the entropy. In this example, despite predicting the correct class ('8'), the RS-NN demonstrates a low confidence (49.7%) and high entropy (2.1626). This holistic approach provides a more nuanced and reliable understanding of the model's uncertainty.

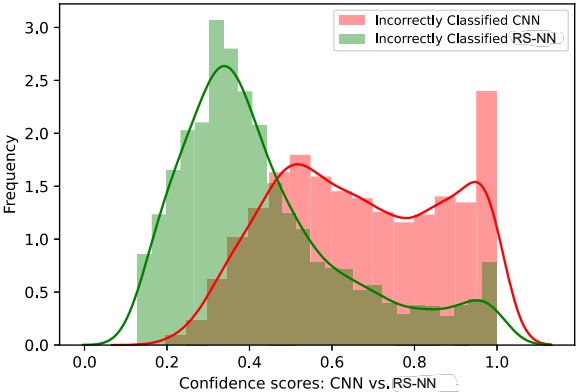

Figure 14: Confidence scores for *Incorrectly Classified samples* of RS-NN and CNN

A distribution of confidence scores for RS-NN in the same noisy experiment are shown in Fig. 15 for both incorrectly classified and correctly classified samples. In Fig. 14, we show how a standard CNN has high confidence for incorrectly classified samples whereas RS-NN exhibits lower confidence and higher entropy (Fig. 16) for incorrectly classified samples. Fig. 16 displays the entropy distribution for correctly and incorrectly classified samples of CIFAR-10, including in-distribution, noisy, and rotated images. Higher entropy is observed for incorrectly classified predictions across all three cases.

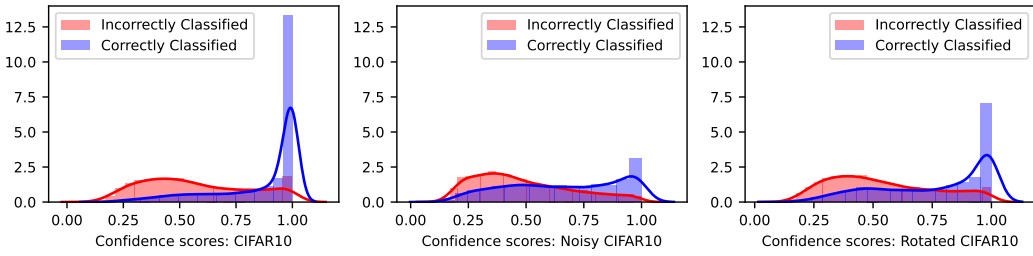

Figure 15: Confidence scores of RS-NN on CIFAR-10, Noisy CIFAR-10, and Rotated CIFAR-10

### E.3 ROBUSTNESS TO NOISY AND ROTATED DATA

We split the MNIST data into training and test sets, train the RS-NN model using the training test, and test the model on noisy and rotated out-of-distribution test data. This is done by adding random noise to the test set of images to obtain noisy data and rotating MNIST images with random degrees of rotation between 0 and 360.

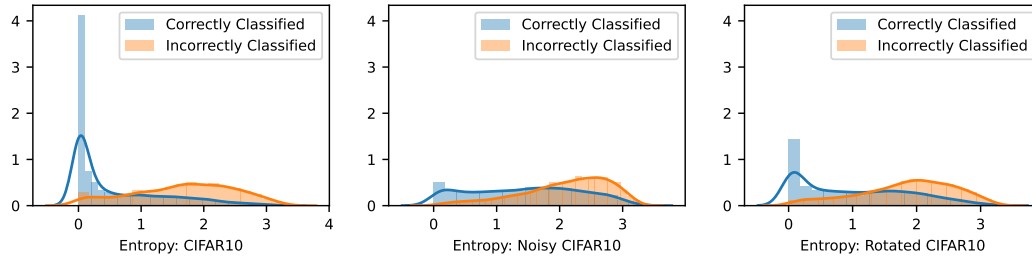

Figure 16: Entropy distribution of RS-NN on CIFAR-10, Noisy CIFAR-10, and Rotated CIFAR-10

Table 10: Standard CNN fails to perform well when tested on noisy noisy and rotated samples of MNIST test data. The predicted results show how a standard CNN predicts the wrong class with a high confidence score, whereas the RS-NN model predicts the right class with varying confidence scores.

| | Standard CNN Predictions | Belief RS-NN Predictions | | |
|---|---|---|---|---|
| **True Label = 2** | | | | |
| | Class 0 **0.628** | **Belief values** | **Mass values** | **Pignistic** |
| | Class 2 0.232 | {'2'} 0.985 | {'2'} 0.982 | **2 0.982** |
| | Class 3 0.117 | {'2', '0', '1'} 0.981 | {'0', '8'} 0.009 | 8 0.007 |
| | Class 8 0.015 | {'2', '1'} 0.980 | {'7', '0', '1'} 0.002 | 0 0.100 |
| | | {'2', '4'} 0.979 | {'7', '8'} 0.002 | 3 0.004 |
| **True Label = 3** | | | | |
| | Class 8 **0.969** | **Belief values** | **Mass values** | **Pignistic** |
| | Class 5 0.018 | {'3', '5'} 0.772 | {'3'} 0.361 | **3 0.427** |
| | Class 9 0.006 | {'6', '3'} 0.637 | {'7', '8'} 0.134 | 8 0.205 |
| | Class 2 0.003 | {'6', '3', '5'} 0.620 | {'0', '8'} 0.104 | 5 0.115 |
| | | {'3'} 0.540 | {'8'} 0.084 | 7 0.080 |
| **True Label = 1** | | | | |
| | Class 2 **1.0** | **Belief values** | **Mass values** | **Pignistic** |
| | Class 1 3.39e-08 | {'2', '1'} 0.999 | {'1'} 0.556 | **1 0.566** |
| | Class 6 1.03e-10 | {'1', '9'} 0.958 | {'2'} 0.423 | 2 0.423 |
| | Class 3 4.92e-11 | {'1'} 0.924 | {'1', '9'} 0.020 | 9 0.010 |
| | | {'1', '5'} 0.825 | {'6'} 4.31e-05 | 6 4.31e-05 |
| **True Label = 9** | | | | |
| | Class 5 **0.988** | **Belief values** | **Mass values** | **Pignistic** |
| | Class 2 0.010 | {'7', '5', '9'} 0.831 | {'7', '9'} 0.171 | **9 0.699** |
| | Class 3 0.0004 | {'7', '9'} 0.706 | {'3'} 0.134 | 7 0.023 |
| | Class 7 0.0002 | {'5', '9'} 0.576 | {'9'} 0.132 | 5 0.004 |
| | | {'3', '9'} 0.558 | {'7', '8'} 0.102 | 3 0.001 |

Tab 10 shows predictions for noisy and rotated noisy MNIST samples. In cases where a standard CNN makes wrong predictions with high confidence scores, Random-Set NN manages to predict the correct class with varying confidences verifying that the model is not overcofident in uncertain cases. For example, a noisy sample with true class '3' has a standard CNN prediction of class '8' with 96.9% confidence, while RS-NN predicts the correct class {'3'} with 42.7% confidence. Similarly, for rotated '9', the standard CNN predicts class '5' with 98.8% confidence whereas RS-NN predicts the correct class {'9'} with 69.9% confidence. For a rotated '1', the standard CNN predicts class '2' with 100% confidence.

Tab. 11 shows the test accuracies for standard CNN, RS-NN, LB-BNN and ENN at different scales of random noise. RS-NN shows significantly higher test accuracy than other models as the amount of noise added to the test data increases.

The test accuracies for standard CNN, RS-NN, LB-BNN and ENN on Rotated MNIST images are shown in Tab 12. The samples are randomly rotated every 60 degrees, -180° to -120°, -120° to -60°,

Table 11: Test accuracies(%) for RS-NN, standard CNN, LB-BNN (Bayesian) and ENN (Ensemble) on noisy samples of MNIST . 'Scale' represents the standard deviation of the normal distribution from which random numbers are being generated for random noise.

| Noise (scale) | 0.2 | 0.3 | 0.4 | 0.5 | 0.6 | 0.7 | 0.8 |
|---|---|---|---|---|---|---|---|
| Standard CNN | 93.40% | 79.08% | 79.15% | 58.33% | 40.19% | 28.15% | 28.59% |
| RS-NN | 96.90% | 85.36% | **85.91** % | **68.33** % | **51.46** % | **37.18** % | **38.05** % |
| LB-BNN | **98.47%** | **95.26** % | 80.18% | 61.56% | 43.28% | 31.41% | 24.55 |
| ENN | 97.81% | 90.76% | 75.41% | 58.99% | 45.70% | 36.97% | 31.51 |

Table 12: Test accuracies(%) for RS-NN, standard CNN, LB-BNN (Bayesian) and ENN (Ensemble) on Rotated MNIST out-of-distribution (OoD) samples. Rotation angle is random between the values given.

| Rotation (angle) | -180/-120 | -120/-60 | -60/0 | 0/60 | 60/120 | 120/180 | 0/360 |
|---|---|---|---|---|---|---|---|
| Standard CNN | 36.41% | 21.86% | 74.41% | 80.80% | 23.89% | 37.53% | 45.86% |
| RS-NN | **37.84%** | **23.54%** | **78.44%** | **81.46%** | **26.31%** | **38.48%** | **47.71%** |
| LB-BNN | 37.56% | 20.19% | 75.12% | 77.67% | 23.18% | 37.83% | 46.07 |
| ENN | 36.40% | 19.43% | 71.70% | 78.59% | 18.70% | 36.38% | 44.14 |

-60° to 0°, etc. A fully random rotation between 0° and 360° also shows higher test accuracy for RS-NN at $47.71\%$ when compared to standard CNN with test accuracy $45.86\%$.

### E.4 ROBUSTNESS TO ADVERSARIAL ATTACKS

We conducted experiments on adversarial attacks using the well-known Fast Gradient Sign Method (FGSM) (Goodfellow et al., 2014). FGSM is a popular approach for generating adversarial examples by perturbing input images based on the sign of the gradient of the loss function with respect to the input. Mathematically, the perturbed image $x'$ is computed as

$$x' = x + \epsilon \cdot \text{sign}(\nabla_x J(\theta, x, y)), \tag{11}$$

where $x$ is the original input image, $\epsilon$ (epsilon) is a small scalar representing the magnitude of the perturbation, $J$ is the loss function, $\theta$ are the model parameters, and $y$ is the true label of the input image.

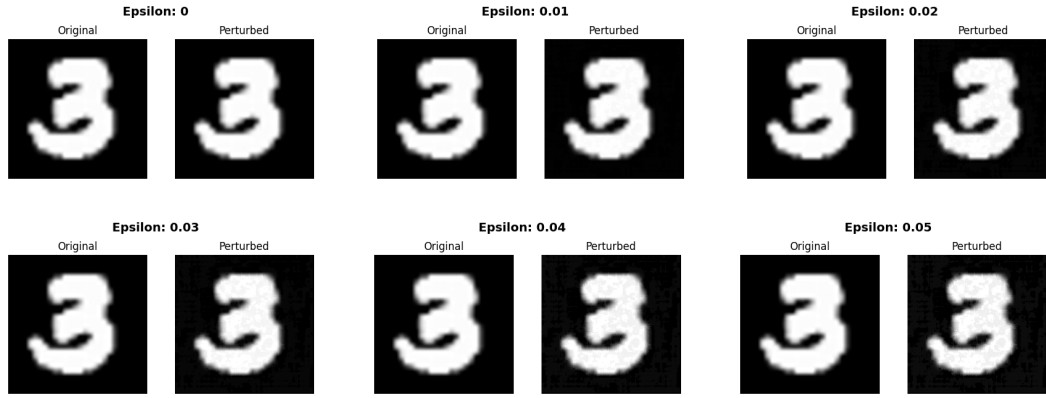

Figure 17: Examples of perturbed images generated using FGSM adversarial attacks on the MNIST dataset for different epsilon values. Epsilon values range from 0 to 0.05.

In our experiments, we applied FGSM adversarial attacks on the MNIST dataset for standard CNN, Random-Set NN (RS-NN), LB-BNN (Bayesian) and ENN (Ensemble) models. This involved perturbing images from the MNIST dataset with noise generated using FGSM based on the gradient of

the loss function (namely, the cross entropy loss for CNN and the $\mathcal{L}_{B-RS}$ loss, see Sec. 3.2, (7), for RS-NN).

Table 13: Test accuracies of CNN, RS-NN, LB-BNN and ENN models under FGSM adversarial attacks on MNIST and CIFAR-10 dataset for different epsilon values. Epsilon values range from 0 to 0.05.

| | Dataset | Model | Epsilon ($\epsilon$) | | | | | | |
|---|---|---|---|---|---|---|---|---|---|
| | | | 0 | 0.005 | 0.01 | 0.02 | 0.03 | 0.04 | 0.05 |
| Test acc. (%) | MNIST | CNN | 99.12 | 98.25 | 94.82 | 72.62 | 49.75 | 38.08 | 32.43 |
| | | RS-NN | **99.71** | 98.46 | 95.84 | 91.90 | **90.62** | **90.10** | **89.72** |
| | | LB-BNN | 99.58 | **99.07** | **98.64** | **97.15** | 80.51 | 47.65 | 34.25 |
| | | ENN | 99.07 | 98.56 | 92.98 | 51.51 | 35.04 | 27.03 | 9.47 |
| | CIFAR-10 | CNN | 92.08 | 41.54 | 20.88 | 9.18 | 5.81 | 4.43 | 4.05 |
| | | RS-NN | **93.53** | **63.42** | **61.73** | **61.67** | **61.53** | **61.12** | **60.35** |
| | | LB-BNN | 89.95 | 23.95 | 10.54 | 6.17 | 5.96 | 6.26 | 6.47 |
| | | ENN | 91.55 | 62.29 | 42.64 | 24.12 | 16.39 | 12.78 | 11.20 |

Subsequently, we evaluated the models' predictions on these perturbed images and computed their test accuracies. Tab. 13 presents the test accuracies of CNN, RS-NN, LB-BNN and ENN models under FGSM adversarial attacks on the MNIST dataset for different values of $\epsilon$.

For CIFAR-10, RS-NN demonstrates higher robustness to the attack compared to all other models with increased perturbations, while for MNIST, LB-BNN has higher accuracy at lower $\epsilon$ values but drops significantly for larger values. RS-NN circumvents the FGSM attack and shows consistent accuracy values for higher $\epsilon$. This is because FGSM for a given input image is dependent on the gradient of the loss, whereas RS-NN makes precise correct predictions with loss/gradient zero and is very confident about these predictions. As a result, the input image does not get perturbed and is correctly classified by RS-NN irrespective of the $\epsilon$ value.

Fig. 17 shows examples of perturbed images generated using FSGM adversarial attacks applied to the MNIST dataset for various epsilon values ranging from 0 to 0.05.

Additionally, we conducted experiments using the Projected Gradient Descent (PGD) adversarial attack (Madry et al., 2017), which is a more iterative and stronger attack compared to FGSM. PGD generates adversarial examples by iteratively perturbing the input image in the direction of the gradient of the loss function, followed by a projection step to ensure that the perturbation stays within a predefined range, typically defined by a norm ball around the original image. Mathematically, the perturbed image $x'$ is computed as follows:

$$x^{(0)} = x, \quad x^{(t+1)} = \text{Proj}_{\mathcal{B}(x,\epsilon)} \left( x^{(t)} + \alpha \cdot \text{sign}(\nabla_x J(\theta, x^{(t)}, y)) \right), \tag{12}$$

where $x^{(t)}$ represents the image at the $t$-th iteration, $\alpha$ is the step size, $\epsilon$ is the maximum allowable perturbation (i.e., the size of the $\mathcal{B}(x, \epsilon)$ ball), and Proj denotes the projection operation that ensures the perturbed image $x'$ stays within the $\epsilon$-ball around the original image $x$:

$$\mathcal{B}(x, \epsilon) = \{x' : \|x' - x\| \leq \epsilon\}. \tag{13}$$

After a number of iterations, typically denoted as $T$, the final perturbed image $x'$ is obtained. This iterative process makes PGD a more powerful adversarial attack compared to FGSM. Tab. 14 presents the test accuracies of CNN, RS-NN, LB-BNN and ENN models under the PGD adversarial attack on MNIST and CIFAR-10 datasets for different values of $\epsilon$.

On MNIST, LB-BNN does well for $\epsilon = 0.005$ and $\epsilon = 0.01$, but CNN, LB-BNN and ENN suffer significant drops in accuracy at higher $\epsilon$ values. RS-NN performs better at higher $\epsilon$ values. On CIFAR-10, RS-NN significantly outperforms all models at all $\epsilon$ values, while the other models show a significant drop in performance, even at $\epsilon = 0.005$. As explained earlier, RS-NN makes precise predictions with gradient zero and, therefore, is unaffected by PGD.

Table 14: Test accuracies of CNN, RS-NN, LB-BNN and ENN models under PGD adversarial attacks on MNIST and CIFAR-10 datasets with $\alpha = 1/255$ and number of iterations = 10. Epsilon values range from 0 to 0.05

| Dataset | Model | Epsilon ($\epsilon$) | | | | | | |
|---|---|---|---|---|---|---|---|---|
| | | 0 | 0.005 | 0.01 | 0.02 | 0.03 | 0.04 | 0.05 |
| | CNN | 99.12 | 98.90 | 55.44 | 14.03 | 8.11 | 6.58 | 6.58 |
| MNIST | RS-NN | **99.71** | 99.13 | 93.39 | **91.06** | **89.25** | **89.07** | **89.02** |
| | LB-BNN | 99.58 | **99.35** | **99.01** | 33.67 | 16.35 | 13.64 | 13.64 |
| Test acc. (%) | ENN | 99.07 | 98.43 | 78.09 | 20.64 | 9.94 | 3.78 | 0.93 |
| | CNN | 92.08 | 23.82 | 2.47 | 0.04 | 0 | 0 | 0 |
| CIFAR-10 | RS-NN | **93.53** | **60.23** | **59.98** | **59.97** | **59.97** | **59.97** | **59.97** |
| | LB-BNN | 89.95 | 6.28 | 0.51 | 0.55 | 0.59 | 0.59 | 0.59 |
| | ENN | 91.55 | 33.95 | 4.40 | 0.15 | 0.02 | 0 | 0 |

## E.5 UNCERTAINTY ESTIMATION FOR RS-NN

In this section, we discuss the uncertainty measures of RS-NN: pignistic entropy (§E.5.1) and credal set width (§E.5.2), and their correlation with confidence scores (§E.5.3).

### E.5.1 ENTROPY OF PIGNISTIC PREDICTIONS

The plots shown in Fig. 18 provide a comparative analysis of the entropy values across various machine learning models under two conditions: In-Distribution (iD) and Out-of-Distribution (OoD) datasets. Each subplot represents a different dataset — CIFAR-10, MNIST, and ImageNet — highlighting the performance of multiple models, including RS-NN, LB-BNN, FSVI, DE, ENN, and CNN.

Fig. 18 illustrates that RS-NN exhibits the best iD vs OoD entropy ratio among the evaluated baseline models. Specifically, RS-NN maintains significantly lower entropy values for iD datasets, indicating a strong ability to recognize familiar patterns while showing elevated entropy for OoD datasets. DE and CNN closely follow RS-NN but struggle to effectively differentiate entropy for the ImageNet dataset. LB-BNN shows more uncertainty regarding iD samples compared to OoD samples. FSVI demonstrates the weakest distinction between iD and OoD.

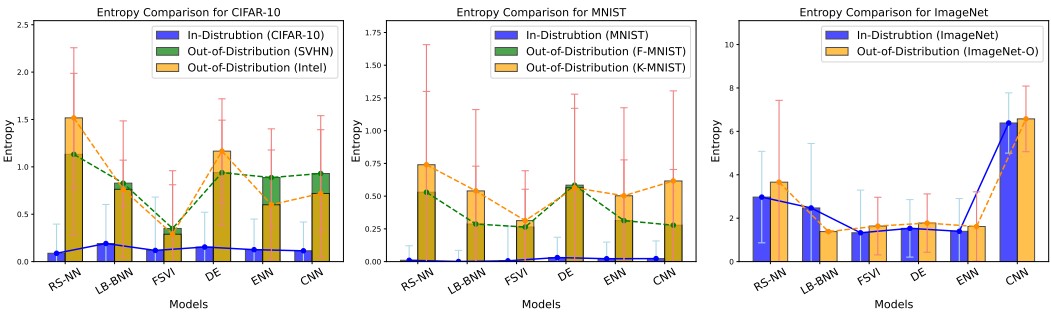

Figure 18: Entropy comparison of all models for iD and OoD datasets. The plots illustrate the performance of various models, with error bars indicating the standard deviation of entropy values.

Tab. 15 shows sample predictions by RS-NN: belief functions, masses, and pignistic predictions for given samples of CIFAR-10. The predicted belief, mass values, pignistic probabilities, and entropy are illustrated for two CIFAR-10 predictions. In the top figure, corresponding to the true label "horse," the model makes a highly confident prediction with 99.9% confidence and a low entropy of 0.0017. Conversely, the bottom figure, associated with the true label "cat," represents an uncertain prediction with 33.3% confidence and a higher entropy of 2.6955. It's important to note that the second image is slightly unclear and poor in quality.

Table 15: The predicted belief, mass values, pignistic probabilities, and entropy for two CIFAR-10 predictions. The figure on the top (True Label = "horse") is a certain prediction with 99.9% confidence and a low entropy of 0.0017, whereas the figure on the bottom (True Label = "cat") is an uncertain prediction with 33.3% confidence and a higher entropy of 2.6955.

| Sample | Belief | | Mass | | Pignistic | | Entropy |
|---|---|---|---|---|---|---|---|
| | {'horse', 'bird'} | 0.9999402 | {'horse'} | 0.9999175 | horse | 0.9998833 | 0.0017040148 |
| | {'horse', 'dog'} | 0.9999225 | {'cat', 'truck'} | 6.859753e-05 | truck | 3.5826409e-05 | |
| | {'horse'} | 0.9999175 | {'ship', 'bird'} | 4.094290e-05 | cat | 3.3859180e-05 | |
| | {'horse', 'deer'} | 0.9998697 | {'horse', 'bird'} | 2.250525e-05 | bird | 3.3015738e-05 | |
| | {'cat', 'truck'} | 7.0380207e-05 | {'dog'} | 1.717869e-05 | ship | 2.3060647e-05 | |
| | {'deer', 'cat'} | 0.4787728 | {'deer'} | 0.3104962 | deer | 0.3332411 | 2.6955228261 |
| | {'deer', 'airplane', 'bird'} | 0.4126398 | {'cat', 'truck'} | 0.1762222 | cat | 0.2230723 | |
| | {'horse', 'deer'} | 0.3732957 | {'dog', 'bird'} | 0.0998060 | horse | 0.1153417 | |
| | {'deer', 'bird'} | 0.3658997 | {'horse', 'bird'} | 0.0954350 | bird | 0.1086245 | |
| | {'deer', 'dog'} | 0.3651531 | {'bird'} | 0.0524873 | dog | 0.1039505 | |

Fig. 19 shows the entropy for two samples of CIFAR-10, one is a slightly uncertain image, whereas the other is certain with high belief values and lower entropy. Both results are shown for $K = 20$ classes, additional to the 10 singletons.

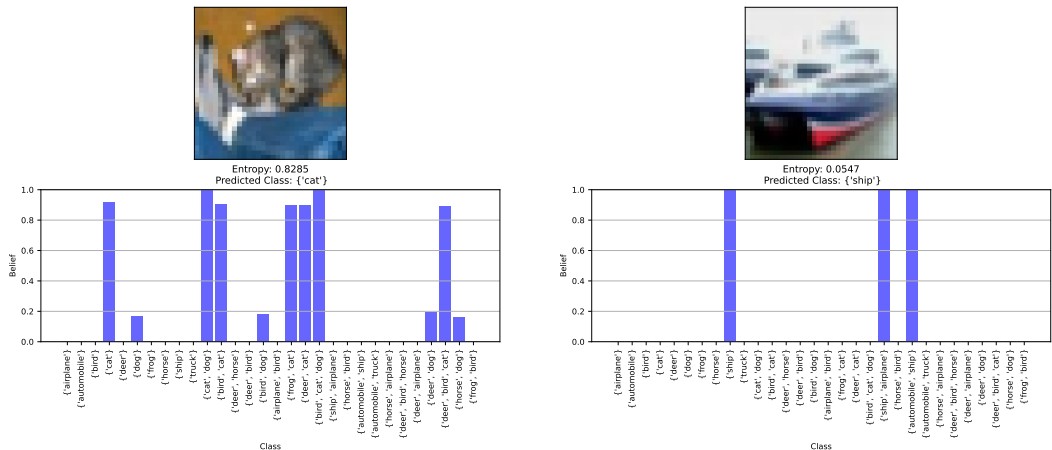

Figure 19: Entropy, predicted class and belief value graph for two samples of CIFAR-10 dataset.

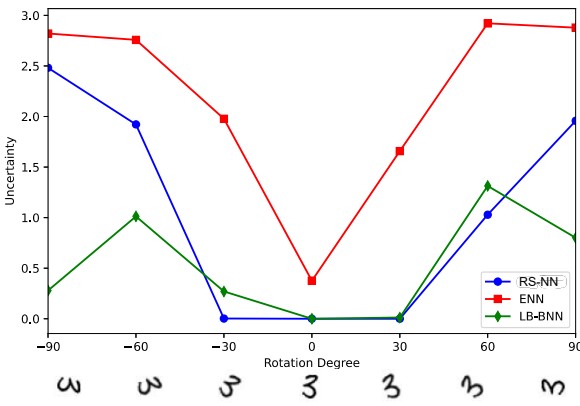

Figure 20: Uncertainty (entropy) of MNIST digit '3' rotated between -90 and 90 degrees. The blue line indicates RS-NN estimates low uncertainty between -30 to 30, and high uncertainty for further rotations.

**Qualitative results of entropy.** Fig. 20 depicts entropy-based uncertainty estimates for rotated out-of-distribution (OoD) MNIST digit '3' samples. All models accurately predict the true class at $-30°$, $0°$, and $30°$. RS-NN and LB-BNN consistently exhibit low entropy in these scenarios,

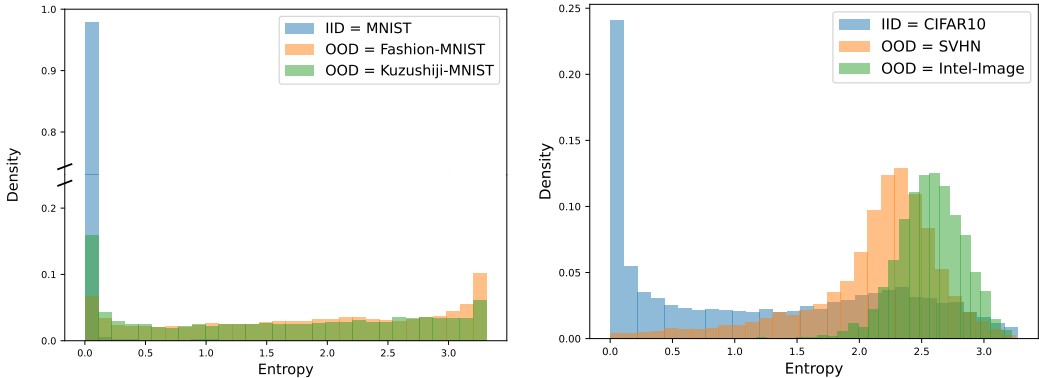

Figure 21: Entropy Distributions for RS-NN on MNIST vs Fashion-MNIST/Kuzushiji-MNIST.

Figure 22: Entropy Distributions for RS-NN on CIFAR-10 vs SVHN/Intel-Image.

while ENN shows higher entropy for correct classifications at these angles. As the rotation angle increases, indicating more challenging scenarios, all models predict the wrong class. Notably, LB-BNN fails to exhibit a significant increase in entropy for these incorrect predictions. ENN performs relatively better than RS-NN at $60°$ and $90°$ rotations but has previously demonstrated high entropy levels even for accurate predictions at relatively minor rotations of $-30°$ and $+30°$ degrees. Overall, RS-NN provides a more reliable measure of uncertainty.

Figs. 21 and 22 show the entropy distributions for RS-NN on MNIST (iD) vs Fashion-MNIST (OoD)/Kuzushiji-MNIST (OoD), and CIFAR-10 (iD) vs SVHN (OoD)/Intel-Image (OoD), respectively. There is a clear iD vs OoD shift in entropy for RS-NN, as detailed in Sec. 4.4.

The plots show that RS-NN consistently exhibits larger iD vs OoD entropy ratio for all datasets. For iD data, the entropy values are generally lower, indicating that these models exhibit a higher level of confidence when making predictions within familiar datasets. Conversely, the OoD side shows significantly higher entropy values for most models, particularly notable with the SVHN and Intel datasets, suggesting that the models struggle to generalize when faced with unfamiliar data. The error bars represent the standard deviation of the entropy values.

### E.5.2 CREDAL SET WIDTH

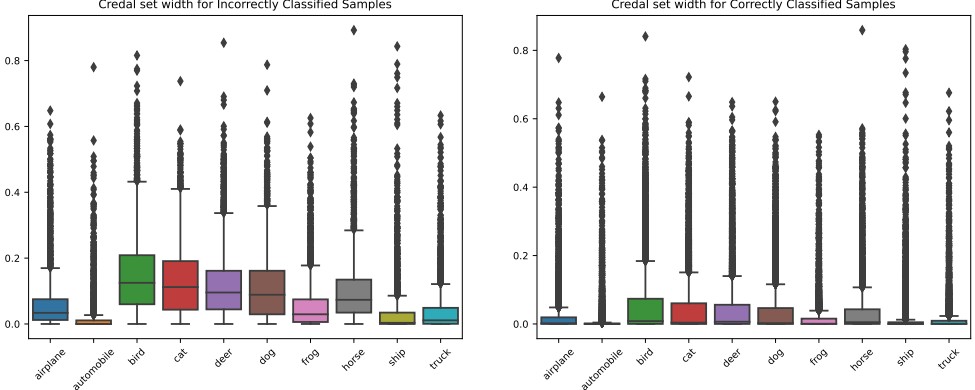

Figure 23: Credal set widths for individual classes of CIFAR-10 dataset. For Incorrectly Classified samples *(top)*. For Correctly Classified samples *(bottom)*.

Fig. 23 plots the credal set widths for incorrectly classified (left) and correctly classified (right) samples over each class $c$ in CIFAR-10, respectively. The box represents the interquartile range between the 25th and 75th percentiles of the data encompassing most of the data. The vertical line inside the box represents the median and the whiskers extend from the box to the minimum and maximum values within a certain range, and data points beyond this range are considered outliers

and are plotted individually as dots or circles. The box plots here are shown for 10,000 samples of CIFAR-10 dataset where most samples within the incorrect classifications have higher credal set widths indicating higher uncertainty in these samples, especially for classes 'bird', 'cat', 'deer', and 'dog'.

### E.5.3 ENTROPY VS CREDAL SET WIDTH

There has been considerable research on uncertainty measures for credal sets. One of the earlier works by Yager (2008) makes the distinction between two types of uncertainty within a credal set: conflict (also known as randomness or discord) and non-specificity. Non-specificity essentially varies with the size of the credal set (Kolmogorov, 1965). Since by definition, aleatoric uncertainty refers to the inherent randomness or variability in the data, while epistemic uncertainty relates to the lack of knowledge or information about the system (Hüllermeier & Waegeman, 2021), conflict and non-specificity directly parallel these concepts. These measures of uncertainty are axiomatically justified (Bronevich & Klir, 2008).

Figs. 24 and 25 depict the relationship between the entropy of RS-NN pignistic prediction and the associated confidence level (Fig. 24), and between credal set width and confidence (Fig. 25) for the CIFAR-10 (left) and SVHN/Intel Image (right) datasets. The distribution of iD predictions (left) for both tests show a concentration at the top left, indicating high confidence and low entropy or credal set width. Conversely, OoD predictions (middle, right) exhibit a more dispersed pattern.

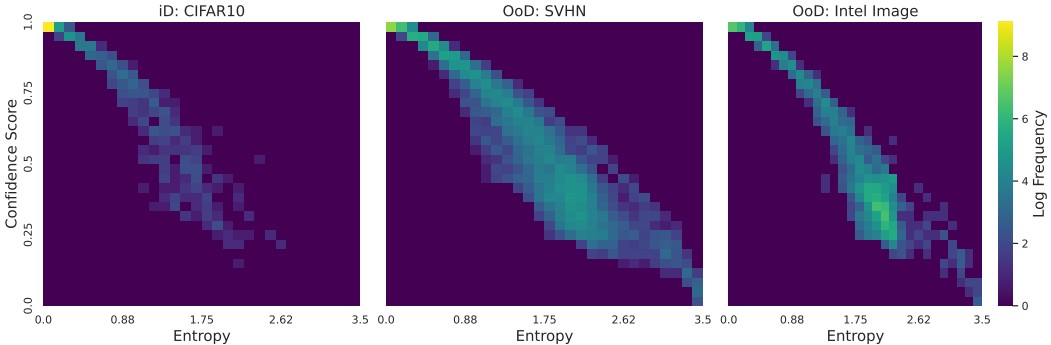

Figure 24: Entropy vs Confidence score on iD (left) vs OoD (right) datasets. For CIFAR-10, most predictions are concentrated top left of the plot indicating lower entropy and higher confidence in the predictions. For SVHN and Intel Image datasets, predictions are more distributed.

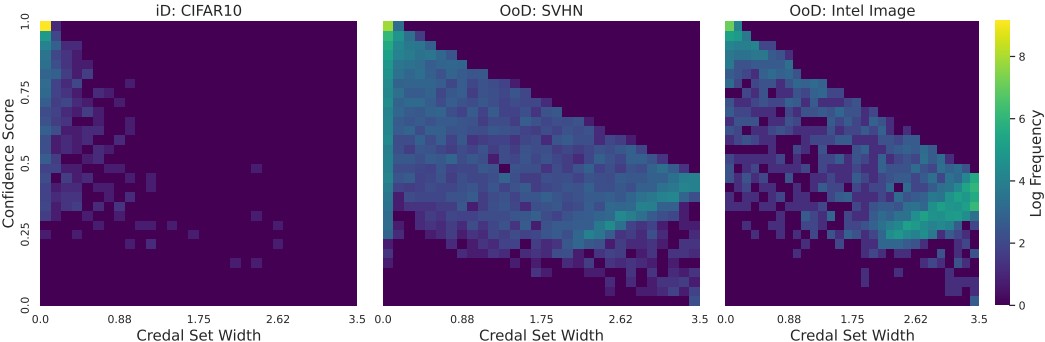

Figure 25: Credal Set Width vs Confidence score on iD (left) vs OoD (right) datasets. For CIFAR-10, confidence scores are high and credal set width is small. For SVHN and Intel Image datasets, credal set width varies for each prediction and is less reliant on confidence score.

Entropy, reflecting prediction uncertainty, is quite correlated with confidence, in both iD and OoD tests. In contrast, as it considers the entire set of plausible outcomes within a belief function rather than a single prediction, credal set width better quantifies the degree of epistemic uncertainty inherent to a prediction. As a result, credal set width is less dependent on the concentration of predictions

and is more reflective of the overall uncertainty encompassed by the model. Fig. 25 shows that, unlike entropy, credal with is clearly not correlated with confidence.

### E.6 ABLATION STUDY ON HYPERPARAMETERS $\alpha$ AND $\beta$

The regularization serves the purpose of ensuring that our model generates valid belief functions, which are constrained by maintaining a sum of masses equal to 1 and ensuring non-negativity. These terms are designed to penalize deviations from valid belief functions. However, it is crucial not to assign too much weight to this term, as excessively penalizing deviations may hinder the model's ability to accurately classify data points. For example, in a Variational Auto Encoder (VAE), if we assign too much weight to the KL divergence term in the loss, the model may prioritize fitting the latent distribution at the expense of reconstructing the input data accurately. This imbalance can lead to poor reconstruction quality and suboptimal performance on downstream tasks.

Hence, it is essential to properly tune the values of $\alpha$ and $\beta$ to ensure that these regularization terms play a meaningful role in the training. The ablation study on hyperparameters is conducted to examine the impact of $\alpha$ and $\beta$ on the model's accuracy. It demonstrates that small values suffice for these parameters.

In Fig. 26, we show the test accuracies for different values of hyperparameters $\alpha$ and $\beta$ in $\mathcal{L}_{B-RS}$ loss function (Eq. 7). Hyperparameters $\alpha$ and $\beta$ adjusts the relative significance of the regularization terms $M_r$ and $M_s$ respectively in $\mathcal{L}_{B-RS}$ loss. The test accuracies are calculated for CIFAR-10 dataset with a fixed number of focal sets $K = 20$ and varying $\alpha$ (blue) and $\beta$ (red) values, $\alpha/\beta =$ [0.001, 0.005, 0.006, 0.009, 0.01, 0.015, 0.020, 0.025, 0.03, 0.04, 0.05]. Test accuracy is the highest when $\alpha$ and $\beta$ equals $1e - 3$.

In our experiments across various datasets and architectures, we found that a value of $1e - 3$ for the hyperparameters yields satisfactory results. This includes architectures ranging from ResNet-50 to Vision Transformers (ViT-Base-16), and datasets ranging from MNIST ($\approx$ 60,000 images of 10 classes) to ImageNet ($\approx$ 1.1M images of 1000 classes). While conducting a parameter search for each dataset could potentially lead to further optimization, it is worth noting that, in many cases, this step can be omitted without sacrificing performance significantly. For further optimization one can, of course, perform a tailored parameter search per dataset, but this is not quite necessary.

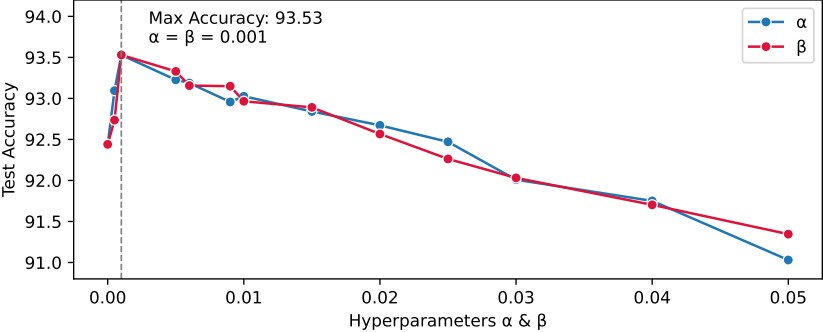

Figure 26: Test accuracies of RS-NN on the CIFAR-10 dataset using ResNet50, for a fixed $K = 20$ and different values of hyperparameters $\alpha$ and $\beta$ in the loss function.

**An alternative loss with valid belief functions.** A straightforward way of enforcing the positivity of the masses is to apply softmax over the masses computed in the BCE loss. This ensures that the masses are non-negative and sum to one. After obtaining the softmaxed masses, one can compute the belief functions from these normalized masses and minimize the loss between this computed belief and the original unnormalized belief.

The random set loss $\mathcal{L}_{RS}$ now becomes,

$$\mathcal{L}_{RS} = \mathcal{L}_{BCE} + \mathcal{L}_{BCE_{norm}},$$

where $\mathcal{L}_{BCE}$ is the BCE loss between belief function logits and ground truth, and $\mathcal{L}_{BCE_{norm}}$ is the BCE loss between the normalized belief function logits reconstructed from mass logits that have been softmaxed.

We implemented this simple method on ResNet50 RS-NN on the CIFAR-10 dataset and obtained an accuracy of $92.47\%$. This is quite close to our original accuracy of $93.53\%$ with the mass regularizations. This mirrors results in the literature showing that soft constraints work as well as hard ones in practice (Márquez-Neila et al., 2017).

### E.7 Ablation Study on the number of focal sets

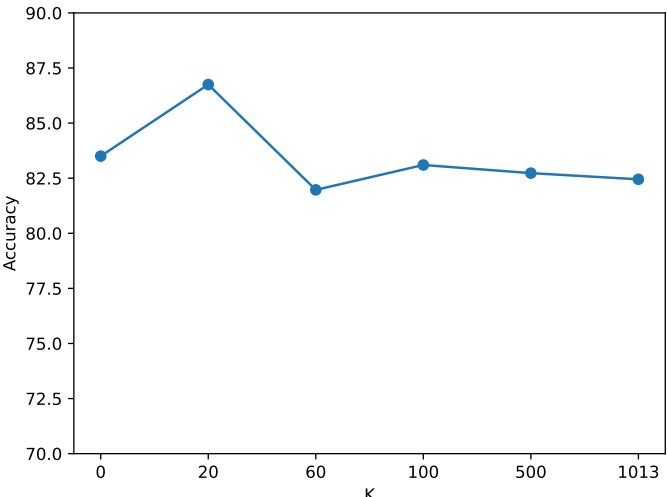

Figure 27: Ablation study on number of non-singleton focal sets $K$ on CIFAR-10 dataset using ViT-Base-16 (with $\alpha = \beta = 1e - 3$). The maximum value of K can be 1013 for 10 classes (after excluding the singletons and empty set).

The number of non-singleton focal sets $K$ to be budgeted is a hyperparameter and needs to be studied. A smaller value of $K$ can lead to more similar results to classical classification while a larger value of $K$ can increase the complexity. Therefore, we conducted an ablation study on $K$ on the CIFAR-10 dataset. We found that a small value of $K$ (comparable to the number of classes) works best and even performs better than $K = 0$ (when there are no non-singleton focal sets) (see Fig. 27). Hence, using set prediction along with the proposed budgeting algorithm not only helps induce uncertainty quantification but also improves the performance of the model. Note that performance comparison was made in terms of accuracy. Smet's Pignistic Transform (Smets, 2005) was used to compute class-wise probabilities from the predicted belief function. Fig. 27 shows an ablation study on different number of focal sets $K$ for CIFAR-10 on the Vision-Transformer (ViT-Base-16) model with $\alpha = \beta = 1e - 3$.

To further support our claim, we compare a budgeted RS-NN with standard RS-NN (full $2^N$ sets) on the CIFAR-10 datasets using ResNet50 as the backbone. In Tab. 16 below, we report the test accuracy, OoD detection metrics (AUROC, AUPRC), pignistic entropy and credal set width for the standard RS-NN vs budgeted RS-NN. The standard RS-NN has 1024 ($2^{10}$) sets since CIFAR-10 has 10 classes, and the budgeted RS-NN has $K = 20$ focal (non-singleton) sets, so $10 + 20 = 30$ input sets. Tab. 16 shows that budgeted RS-NN performs better than the standard model, especially iD vs OoD entropy, and AUROC and AUPRC scores.

As we say in the main paper, while, in theory, using the full power set is ideal for uncertainty estimation, in practice, having all those degrees of freedom might make it more difficult for the network to learn. When a model is confronted with an overwhelming number of input sets, it may struggle to identify meaningful patterns amidst the noise created by less relevant or redundant combinations. This excessive flexibility can hinder the network's ability to converge effectively, as

it may become trapped in local minima or overfit to the training data. This can dilute the model's capacity to generalize well to unseen data, thereby complicating the learning of the underlying relationships between features and the set structure of the data.

Table 16: Comparison of accuracy, AUROC, AUPRC, pignistic entropy, and credal set width for standard RS-NN vs budgeted RS-NN on the CIFAR-10 dataset.

| Dataset | Model | In-distribution (iD) | | Out-of-distribution (OoD) | | | |
| | | Test accuracy (%) (↑) | ECE (↓) | SVHN | | Intel Image | |
| | | | | AUROC (↑) | AUPRC (↑) | AUROC (↑) | AUPRC (↑) |
| CIFAR-10 | Standard RS-NN | 92.66 | 0.0501 | 93.39 | 91.38 | 95.84 | 89.33 |
| | Budgeted RS-NN | **93.53** | **0.0484** | **94.91** | **93.72** | **97.39** | **90.27** |
| | | Entropy (iD) (↓) | Credal Set Width (iD) (↓) | SVHN | | Intel Image | |
| | | | | Entropy (↑) | Credal Set Width (↑) | Entropy (↑) | Credal Set Width (↑) |
| | Standard RS-NN | $0.286 \pm 0.80$ | $0.048 \pm 0.15$ | $\mathbf{1.205 \pm 1.20}$ | $\mathbf{0.408 \pm 0.07}$ | $1.490 \pm 0.71$ | $\mathbf{0.669 \pm 0.21}$ |
| | Budgeted RS-NN | $\mathbf{0.088 \pm 0.308}$ | $\mathbf{0.007 \pm 0.044}$ | $1.132 \pm 0.855$ | $0.260 \pm 0.322$ | $\mathbf{1.517 \pm 0.740}$ | $0.587 \pm 0.367$ |

### E.8 ON THE FEASIBILITY OF BUDGETING OF SETS

Tabs. 17 and 18 show that replacing t-SNE with UMAP for budgeting in the RS-NN model yields comparable performance in terms of test accuracy, out-of-distribution (OoD) detection, and uncertainty estimation. Both t-SNE and UMAP are capable of embedding data in a 3D space, with UMAP employing a faster algorithm that utilizes five nearest neighbors and a minimum distance of 0.9. This approach ensures that the integrity of the data structure is maintained while significantly reducing computational costs. UMAP offers an efficient alternative without compromising the model's effectiveness. To make the generation of embeddings for budgeting of focal sets more efficient, we can utilize faster alternatives like UMAP instead of t-SNE, without compromising the overall model performance.

Table 17: Test accuracy and OoD detection (AUROC, AUPRC) results on RS-NN where budgeting is done using t-SNE vs budgeting done on UMAP.

| Dataset | Model | In-distribution (iD) | | Out-of-distribution (OoD) | | | |
| | | Test accuracy (%) (↑) | ECE (↓) | SVHN | | Intel Image | |
| | | | | AUROC (↑) | AUPRC (↑) | AUROC (↑) | AUPRC (↑) |
| CIFAR-10 | RS-NN (t-SNE) | 93.53 | 0.0484 | 94.91 | 93.72 | 97.39 | 90.27 |
| | RS-NN (UMAP) | 92.97 | 0.0482 | 94.31 | 92.46 | 96.22 | 89.66 |

Table 18: Entropy and credal set width results on RS-NN where budgeting is done using t-SNE vs budgeting done on UMAP. The goal is to minimize both metrics for in-distribution (iD) data while increasing them for out-of-distribution (OoD) data.

| Dataset | Model | In-distribution (iD) | | Out-of-distribution (OoD) | | | |
| | | Entropy (↓) | Credal Set Width (↓) | SVHN | | Intel Image | |
| | | | | Entropy (↑) | Credal Set Width (↑) | Entropy (↑) | Credal Set Width (↑) |
| CIFAR-10 | RS-NN (t-SNE) | $0.088 \pm 0.308$ | $0.007 \pm 0.044$ | $1.132 \pm 0.855$ | $0.260 \pm 0.322$ | $1.517 \pm 0.740$ | $0.587 \pm 0.367$ |
| | RS-NN (UMAP) | $0.082 \pm 0.305$ | $0.007 \pm 0.047$ | $1.119 \pm 0.887$ | $0.341 \pm 0.374$ | $1.423 \pm 0.762$ | $0.582 \pm 0.402$ |

A 2D visualization of the ellipses formed by calculating the principle axes and their lengths using eigenvectors and eigenvalues obtained from GMMs, for each class $c$ (Spruyt, 2013) is shown in Fig. 28.

The budgeting procedure can also be modified to accommodate multiple data sources or a continuous stream of data. t-SNE can be replaced by an autoencoder (this will need to be trained first) or PCA to add continual inference capabilities in the dimensionality reduction step. Similarly, Gaussian-based dynamic probabilistic clustering (GDPC) (Diaz-Rozo et al., 2018) can be used instead of GMM clustering to handle a continuous stream of data. GDPC works by first initialising a GMM and then updating its parameters as it sees new data. In alternative, (Kulis & Jordan, 2011) have proposed an interesting approach to learn a GMM from multiple sources of data by maintaining a local and a global mixture model. Both these approaches can be plugged into our budgeting framework with

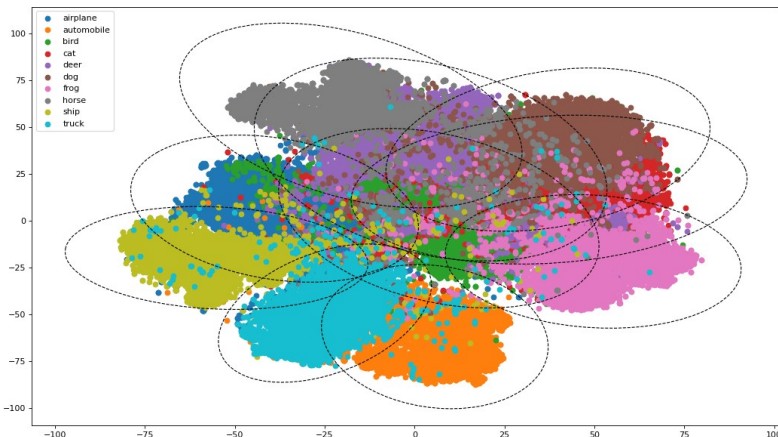

Figure 28: 2D visualization of the clusters of 10 classes of CIFAR-10 dataset and the ellipses formed by RS-NN based on the hyperparameters of Gaussian Mixture Models.

little modifications to add continual learning capabilities to it. The computation of cluster overlaps remains the same, so overlapping scores and the resultant focal sets will need to be updated as the clusters evolve. However, a cluster tracking strategy (Barbara & Chen, 2001) can be employed so that the overlap assessment step is only re-done when a sufficient drift has been detected in the clusters. All the proposed changes are efficient enough to not have a significant effect on the overall time of budgeting.

## F    APPLICATION ON TEXT CLASSIFICATION

The random-set approach can be applied on any classification task. In this section, we detail experiments performed on text classification.

**Dataset.** We chose the BBC text dataset (Greene & Cunningham, 2006), which consists of various news categories such as tech, business, sport, and entertainment, and used this data for training our model. The task is to classify the given text into one of these categories. The dataset is structured with two columns: category and text, where the category is the label, and the text is the content to be classified. The text data will be preprocessed and fed into the model to predict the category of each text.

**Model.** To train our model, we leveraged a pre-trained BERT model available on TensorFlow Hub. Specifically, we used the small BERT ($L - 4_H - 512_A - 8$) model, which is a lightweight version of BERT. The model is designed to take text input, preprocess it with BERT's preprocessing layer, pass it through BERT's encoder to generate embeddings, and then use a dropout layer. The final layer is a fully-connected layer with sigmoid activation for the RS-NN BERT classifier (RS-NN BERT), and a fully-connected layer with softmax activation for the standard BERT classifier (CNN BERT).

**Training.** We trained these models on the BBC text dataset, fine-tuning the BERT model as part of the training process. Both models were trained for 10 epochs using Adam optimizer, a batch size of 32, and a learning rate of 3e-5. RS-NN BERT has all the sets of classes excluding the null set and full set ($2^5$ = 32 - 2 = 30 classes) and CNN BERT has the 5 original classes.

**Out-of-distribution detection.** For OoD detection, we use the Emotion Detection from Text (Seyeditabari et al., 2018) dataset which contains tweets annotated with emotional labels. The dataset includes three columns: tweetid, sentiment, and content, where sentiment represents the emotion behind each tweet. The dataset includes 13 emotion classes such as anger, fear, joy, love, sadness, and surprise, etc, aiming to identify emotional expressions in text.

**Experimental results.** In Tab. 19 below, we show the test accuracy, out-of-distribution (OoD) metrics (AUROC, AUPRC), and the in-distribution (iD) vs OoD entropy for RS-NN BERT and CNN BERT. RS-NN achieves higher overall performance with a significantly higher AUROC and AUPRC scores, highlighting the efficiency of the model at differentiating between iD and OoD samples. RS-NN has higher OoD entropy than CNN, but also has higher iD entropy than CNN, which indicates that the uncertainty in predictions is higher as it is a small dataset.

Table 19: Test accuracy, AUROC, AUPRC, iD vs OoD entropy for RS-NN BERT and CNN BERT (both fine-tuned on BERT).

| Dataset (iD) | Model | Test Acc. (%) ($\uparrow$) | iD Entropy ($\downarrow$) | Emotion Detection (OoD) | | |
| --- | --- | --- | --- | --- | --- | --- |
| | | | | AUROC ($\uparrow$) | AUPRC ($\uparrow$) | Entropy ($\uparrow$) |
| BBC Text (iD) | RS-NN BERT | **96.85** | $0.541 \pm 0.196$ | **95.19** | **95.93** | $\mathbf{1.510 \pm 0.611}$ |
| | CNN BERT | 94.15 | $\mathbf{0.027 \pm 0.125}$ | 56.71 | 57.54 | $0.059 \pm 0.191$ |

