# OpenReview forum: "Random-Set Neural Networks"
_ICLR.cc/2025/Conference — ICLR 2025 Poster_

### Official Review · Reviewer_N6Y1 · 2024-10-30

**Soundness:** 2
**Presentation:** 4
**Contribution:** 2
**Rating:** 5
**Confidence:** 4

**Summary:**

This paper introduces Random-Set Neural Networks (RS-NN), a novel approach to classification that predicts belief functions over sets of classes rather than probability vectors. The key innovation is using random sets and belief functions to model epistemic uncertainty arising from limited training data. RS-NN extends any baseline neural network by modifying its output layer to predict belief values for relevant sets of classes, identified through an efficient "budgeting" procedure that selects the most informative class combinations. The approach measures uncertainty through both the entropy of the "pignistic" (central) probability prediction and the width of the predicted credal set (set of compatible probability distributions). Through extensive experiments on multiple benchmarks, the authors claim RS-NN outperforms state-of-the-art Bayesian and ensemble methods in accuracy, out-of-distribution detection, uncertainty quantification, and adversarial robustness, while being computationally efficient. The approach is applied to different network architectures and provides statistical guarantees through conformal prediction.

**Strengths:**

This paper describes a method in uncertainty estimation for deep learning, using the Random-Set Neural Network (RS-NN). The work discusses a theoretical framework using random sets and belief functions to model epistemic uncertainty, develops an innovative "budgeting" technique to make the approach computationally tractable, and proposes new uncertainty measures based on pignistic entropy and credal set width. The paper presents comprehensive empirical validation across multiple datasets and architectures showing improved performance to state-of-the-art Bayesian and ensemble methods in accuracy, out-of-distribution detection, and uncertainty quantification. The presentation is clear, with explanations of complex concepts supported by helpful visualizations and detailed appendices. The work is significant for machine learning deployment in safety-critical domains, offering a wrapper technique applicable to existing architectures while providing uncertainty estimates and robustness to adversarial attacks. The theoretical foundations in random set theory, combined with practical engineering solutions like the budgeting procedure, make this work both mathematically rigorous and implementable in practice.

**Weaknesses:**

The authors claim in the abstract "In this paper, we propose a novel Random-Set Neural Network (RS-NN) approach to classification
which predicts belief functions" and go on to cite Manchingal & Cuzzolin, 2022 (w.r.t. epistemic uncertainty) where on pg 18 it is "discussed under Sec. 6, the performance and evaluation metrics for random set neural networks", as well as several mentions of belief functions throughout the earlier paper. The works seem closely related hence the claim to originality in the more recent work seems questionable, and a clarification in order.

**Questions:**

1. Please address the point raised in section Weaknesses.
2. The authors claim "Current machine learning models struggle to provide reliable predictions when confronted with unfamiliar data (Guo et al., 2017a)". The paper is from 2017. How did the authors infer no progress has been made since ?
3. The authors claim "RS-NN uses a random-set framework which does not require prior assumptions and is more data driven". Both models are classifying the same data, hence the claim that one is more data driven than the other does not hold and additional details are required to make the text clear.
4. The authors provided code to replicate results. When running eval.ipynb, line:

model = load_model(selected_model, selected_dataset, model_type = "CNN")

threw error:

OSError: No file or directory found at saved_models/CNN_resnet50_cifar10.keras

The weights file was found to be missing. Can the file be provided ?

---

### Official Review · Reviewer_KLiR · 2024-10-31

**Soundness:** 3
**Presentation:** 4
**Contribution:** 4
**Rating:** 8
**Confidence:** 4

**Summary:**

The authors introduce a novel Random-Set Neural Network (RS-NN) approach for epistemic uncertainty estimation in classification. RS-NN can be applied on top of an existing neural network and predicts distributions (belief functions) over collections of sets of classes and makes use of pignistic prediction to estimate the entropy. The authors theoretically motive their work and empirically show, that RS-NN outperforms state-of-the-art Bayesian and Ensemble uncertainty estimation methods on multiple metrics and datasets.

**Strengths:**

- The manuscript is generally well written and in most parts easy to follow. Examples and visualizations are provided where necessary or helpful for the reader.
- The authors deliver a good example for belief functions in Section 2 which easy to follow, and greatly helps to understand the previous definitions.
- Visualizations in Figure 1, 3, 4 and 5 are well crafted and complement the understanding of the described procedures.
- The experimental evaluation is detailed and covers multiple methods, datasets and aspects to investigate. Most experiments were run multiple times, the standard deviation was reported in addition to the mean result.
- The proposed method impressively beats most of the previous state-of-the-art results in terms of accuracy, uncertainty estimation, and out-of-distribution detection while keeping the inference time of the base model (note, that this does not include the additional time one-time cost for the budgeting step).
- The appendix contains a lot of additional experiments and in-detail explanations and recaps for relevant methods. While being relatively long (23 pages), the authors managed to insert references to the appendix in the main body of the manuscript at many relevant places. I found myself looking into the appendix more than usual compared to other papers with long appendices (which I see as positive).

**Weaknesses:**

The experimental results solely focus on image classification tasks. What about a different modality? Tabular data? Text data? The proposed method is not specific to image data and can be applied to arbitrary models and modalities with classification tasks.

Not having worked with belief functions yet, it was at first unclear to me, that Eq. (1) goes over *all possible subsets A* since L185 talks about "mass values to *its* subsets", implying that $m(\cdot)$ is a function which is assigned / is only valid on its own set of subsets. Suggestion: make this consistent with the use of the powerset notation used in the directly following example (L193) and replace $\forall A \subset \Theta$ in the middle of Eq. (1) with $A \in \mathbb{P}(\Theta)$ at the end of the equation.


Minor Errors and Suggestions (these did not impact the score):
- L30: "how confident ti" -> "how confident it"
- L100: "can applied" -> "can be applied"
- L103: "we outline an **ingenious** budgeting method" -> In my opinion, the authors should refrain from calling their own contributions "ingenious".
- L183 - L187: This part is a bit confusing, stating "While [...] mass functions does X, a belief function does Y:", then following with an equation of the mass function (the sentence implies, that the reader will now be shown the definition of a belief function, which only comes in the equation after the next paragraph).
- L269 and multiple times in the appendix: Wrong quotation marks around words were used ("foobar" instead of ``foobar'').
- L305: The definition of $M_r$ should appear directly after introducing $M_r$ in the text. I guess the authors squashed this into Eq. (6) together with $M_s$ to safe space, but it would improve readability to separate both definition, since right now the text talks about $M_s$ ("we also add a mass sum term $M_s$, namely:") and then shows a definition of $M_s$ *and* $M_r$. Since the camera-ready version does not allow an additional page, the authors could also rewrite the text preceding Eq. (6) to make this more clear.
- L432, Tab. 2: The highlighted results (bold-face) for "In-distribution Entropy" for ENN on ImageNet is wrong and should be FSVI (lowest), and for "Out-of-distribution Entropy" for RS-NN on F-MNIST is wrong and should be DE (highest).
- L1362: "to a 95The class" -> "to a 95% confidence interval. The class" (?) -- I guess the "%" was not escaped and commented out the rest of the line in LaTeX.
- L2144, E.7, Tab. 15: The highlighted results (bold-face) for "Entropy (OoD)", "Credal Set Width (OoD)" on SVHN and "Credal Set Width (OoD)" on Intel Image are flipped (in all cases "Budgeted RS-NN" was highlighted as "winner" while "Standard RS-NN" achieved a better performance).
- E.4, L1830: The authors speak of "Tab. 1 in the anonymous PDF linked to this response". This reads like a leftover from a previous submission in which this part of the appendix was a response to a review.

**Questions:**

I have compared the reported results for LB-BNN, FSVI, and Deep Ensemble with their sources and was surprised to find large differences in the reported metrics compared to the results reported in the manuscript in Table 2 (OoD detection performance and uncertainty estimation).

While the manuscript claims "Note that the results in the FSVI paper (Rudner et al., 2022) are based on a pre-trained ResNet18", having read the FSVI paper and supplementary, I was not able to confirm this. See also Appendix D.3 of FSVI[1]:

> In this experiment, a standard ResNet-18 network architecture was used. All models are trained for 200 epochs with a mini-batch size of 128 using SGD with a learning rate of 5 × 10−3, momentum (with momentum parameter 0.9), and a cosine learning rate schedule with parameter 0.05

The reported metrics use ResNet-50 everywhere, while LB-BNN, FSVI, and Deep Ensemble use a ResNet-18 on CIFAR-10. Using ResNet-50 (25.6M parameters) should in principle only improve the results as we increase model capacity, compared to ResNet-18 (11.7M parameters).

**CIFAR-10**

| Method                      | Model                                             | Test accuracy (%) | ECE $\downarrow$ | AUROC (SVHN) $\uparrow$ |
|-----------------------------|---------------------------------------------------|:-------------------:|:------------------:|:-------------------------:|
| RS-NN                       | ResNet-50                                         | 93.53             | 0.0484           | 94.91                   |
| LB-BNN (reported)           | ResNet-50                                         | 89.95             | 0.0585           | 88.14                   |
| LB-BNN (source)             | ResNet-18                                         | **95.40**         | 0.3090           | 96.50                   |
| FSVI (reported)             | ResNet-50                                         | 80.29             | 0.0521           | 80.59                   |
| FSVI (source)               | ResNet-18 ($p_{x_{C}} = \text{rand. monochrome}$) | 93.35             | 0.0340           | 94.76                   |
| FSVI (source)               | ResNet-18 ($p_{x_{C}} = \text{CIFAR-100}$)        | 93.57             | 0.0260           | 98.07                   |
| FSVI Ensemble (source)      | ResNet-18 ($p_{x_{C}} = \text{rand. monochrome}$) | 95.19             | **0.0130**       | **99.19**               |
| Deep Ensemble (reported)    | ResNet-50                                         | 92.73             | 0.0482           | 93.84                   |
| Deep Ensemble (FSVI source) | ResNet-18                                         | 95.13             | 0.0190           | 98.04                   |

I appreciate, that the authors took the time and effort to run the compared methods themselves, improving comparability in principle. Nevertheless, it surprises me, that the original (highlighted as *source* in the table above) results for LB-BNN, FSVI, and DE all *outperform* RS-NN in test accuracy, ECE, and AUROC of OoD detection by a significant margin. I have focused on the CIFAR-10 results, as this was a dataset with SVHN as OoD dataset used in all three sources.

Could it be, that the compared methods are, as always, are sensitive to hyperparameters, and the authors probably have unintentionally picked suboptimal hyperparameters for those methods by accident? Or am I missing some important detail?

While I generally really like the manuscript and experimental evaluation, the above leaves me questioning the validity of the comparisons and the claims of "beating state-of-the-art methods" in all experiments. I usually do not particularly care about beating the state-of-the-art, but if a method happens to be misrepresented and suddenly goes from being the best method to being the worst method in the comparison, I get interested as to why that is (as I said, I'm probably just missing an important detail -- I'm not implying that the authors did misrepresent anything on purpose). I will happily raise my score to 6 or 8 if the authors can clarify these discrepancies.

[1] Tractable Function-Space Variational Inference in Bayesian Neural Networks; Tim G. J. Rudner, Zonghao Chen, Yee Whye Teh, Yarin Gal. https://arxiv.org/pdf/2312.17199

**Misc. Questions**:
- L80: "the entropy of a pignistic prediction is more diverse than that of a Bayesian prediction across in-distribution and out-of-distribution data (see Tab. 2)" -> What does "more diverse" mean?
- E.4, Tab. 13: Why was the influence of $\varepsilon$ not evaluated against LB-BNN and ENN as the authors did in Table 12 for different rotations of MNIST and Table 11 for different perturbation noise levels?
- L212, Eq. (4): How is the set of probability distributions $P$ on $\Theta$ selected/constructed?

---

> ### Comment · Reviewer_KLiR · 2024-11-20
> **All concerns and questions were addressed by the authors.**
>
> I thank the authors for their in-depth responses. Since the authors have addressed **all** of my concerns and questions, I've happily raised my score to 8 and my confidence to 4.
>
> If the authors feel like it, I would recommend adding some variation of the justification for the deviations in the results (the authors results vs. original paper results) in the appendix, simply to ease readers like me who see the results and maybe know the original paper's results and think "hey wait, this looks off". I think all points given by the authors are valid, and it is worth highlighting the effort the authors have put in being as close as possible to the original works setups while keeping it comparable across the board.

---

### Official Review · Reviewer_8ggx · 2024-11-03

**Soundness:** 3
**Presentation:** 3
**Contribution:** 3
**Rating:** 8
**Confidence:** 3

**Summary:**

This paper focuses on a novel approach called Random-Set Neural Network (RS-NN) for classification tasks. The RS-NN approach aims to predict belief functions over a class list using random sets, which can represent uncertainty induced by limited training data. RS-NN can be presented as a wrapper technique that can enhance existing baseline models by predicting beliefs for sets of classes. Specifically, it efficiently selects relevant sets of classes using GMMs and introduces measures for assessing uncertainty in random-set predictions. In the experiments, RS-NN is compared with Bayesian and Ensemble methods, demonstrating better performance in uncertainty estimation on multiple benchmarks.

**Strengths:**

1. The paper is well-written and well-organized.
2. The issue of uncertainty in neural networks is crucial. The proposed method can be easily extended to other baselines, which could benefit the model trustworthiness community.
3. The proposed method is much more efficient than other baselines, making the proposed method more accessible and feasible for adoption.
4. The evaluation is sufficient. The effectiveness of the proposed method is validated on various models and datasets.

**Weaknesses:**

1. One concern lies in the formulation of Eq. 7, where the validity of belief functions is achieved by the involvement of loss terms $M_s$ and $M_r$. However, according to the ablation studies in Figure 26, a large hyperparameter of $\alpha$ and $\beta$ could lead to a decrease in accuracy, which makes it a trade-off between the accuracy and validity of belief functions. How to balance this trade-off is not well-explained in this paper. Although the authors mentioned that any negative masses will be set to zero in post-training, the gap caused by this process is not discussed or explained.
2. Another concern lies in the comparison. Most recent model calibration works that focus on uncertainty have not been discussed or compared in this paper, such as [a,b,c,d]. Furthermore, the evaluation of adversarial robustness could be weak. It is necessary to conduct some evaluation on CIFAR-10 under a stronger PGD attack to demonstrate the robustness.

[a]. Calibrating Deep Neural Networks using Focal Loss. NeurIPS 2020.

[b]. Improving model calibration with accuracy versus uncertainty optimization. NeurIPS 2020.

[c]. Local Calibration: Metrics and Recalibration. UAI 2022.

[d]. Dual focal loss for calibration. ICML 2023.

**Questions:**

1. Please clarify the validity of belief functions in Eq. 7.
2. Please discuss and compare with some recent uncertainty works in the experiments.

---

### Meta-Review · Area_Chair_RtGV · 2024-12-19

**Metareview:**

The paper introduces a novel method to augment neural networks with epistemic uncertainty estimates by providing distributions (belief functions) over the collection of sets of classes. The respective approach is referred to as random-set neural networks (RS-NN). The advantages of RS-NNs are numerous, ranging from its applicability and practical efficiency, to comprehensive evaluation with strong results. Reviewers acknowledged the contributions and thought the paper to be well-structured and well-written. Two main points for requested improvements were to investigate a modality that is not images, and to conduct a deeper investigation with respect to robustness in attack scenarios. The authors have addressed these through additions in the discussion phase to the full satisfaction of two reviewers. The third reviewer’s only raised weakness seems to be a misunderstanding of a lack of novelty because of related work mentioned in the paper. The authors have provided adequate qualification to this point.
Overall, the AC agrees with the reviewers that the paper is well-written, the method important, and its strong results over e.g. ensembles practically useful. The recommends to accept the paper.

**Additional Comments On Reviewer Discussion:**

Two out of three reviewers have provided extensive feedback with several requests for clarifications and additions. The authors have faithfully addressed the feedback points in the discussion and have made several improvements to the paper to address the concerns. One particularly important concern was a clarification on a mismatch of reported results from those reported in a prior work. The authors have provided plausible explanations on why results deviate and why reproducibility of some original work was an issue. These remarks have made it into a new appendix section. The reviewer was satisfied with the clarifications and the AC also believes that the provided answers are reasonable.

Finally, the third reviewer has not engaged in the discussion period. The initial concerns seem to have been light and received a rebuttal from the authors. The mentioned points, e.g. a citation from 2017 that is slightly outdated and a concern over too close a relation to prior work, were easily addressed. The reviewer did unfortunately not acknowledge these clarifications, but the AC believes they are fully justified. Given that the reviewer has not responded and the aspects were addressed, this point has weighed in less strongly in the AC’s final decision.

---

### Decision · Program_Chairs · 2025-01-22

Accept (Poster)